# Spatial-ID: a cell typing method for spatially resolved transcriptomics via transfer learning and spatial embedding

Rongbo Shen [1,7], Lin Liu [2,7], Zihan Wu [1,7], Ying Zhang [2,7], Zhiyuan Yuan [1,3], Junfu Guo [2], Fan Yang[1], Chao Zhang[2], Bichao Chen [2], Wanwan Feng[1,4], Chao Liu[2], Jing Guo[2], Guozhen Fan[2], Yong Zhang[2,5], Yuxiang Li[2,5] ✉, Xun Xu [2,6] ✉ & Jianhua Yao[1] ✉

Spatially resolved transcriptomics provides the opportunity to investigate the gene expression profiles and the spatial context of cells in naive state, but at low transcript detection sensitivity or with limited gene throughput. Comprehensive annotating of cell types in spatially resolved transcriptomics to understand biological processes at the single cell level remains challenging. Here we propose Spatial-ID, a supervision-based cell typing method, that combines the existing knowledge of reference single-cell RNA-seq data and the spatial information of spatially resolved transcriptomics data. We present a series of benchmarking analyses on publicly available spatially resolved transcriptomics datasets, that demonstrate the superiority of Spatial-ID compared with state-of-the-art methods. Besides, we apply Spatial-ID on a self-collected mouse brain hemisphere dataset measured by Stereo-seq, that shows the scalability of Spatial-ID to three-dimensional large field tissues with subcellular spatial resolution.

In the last decade, single-cell RNA-seq (scRNA-seq) technologies have made significant progress toward the systematic characterization of cell dynamics[1,2]. However, the dissociation step of scRNA-seq leads to loss of spatial information, preventing the investigation of tissue organization in its naive state[3]. To reveal fine-scale spatial organization and microenvironments within tissues, various spatially resolved transcriptomics (SRT)[4–6] technologies have been introduced to map gene expression profiles to the spatial context. Early technologies attempted to spatially map multiplexed gene expression by either in situ hybridization (ISH)[7,8] or spatially barcoded oligo-deoxythymidine microarrays[9,10]. More advanced ones, e.g., seqFISH[11], seqFISH+[12], MERFISH[8], Slide-seq[13,14], HDST[15], and Stereo-seq[16], have been proposed to improve in terms of spatial resolution, gene throughput, and transcript detection sensitivity, allowing for comprehensive mapping of transcripts over large tissue sections.

Cell type annotation is a fundamental task for cell and tissue biology[17] that can help characterize the biological process of tissues at single-cell level. This task is conventionally performed by single-cell transcriptome analysis on the data acquired by scRNA-seq technology. Facing the exponential growth of high-dimensional and noisy data associated with sequencing technologies[18], it requires high performance annotation methods that can effectively reduce dimensionality and are robust to data noise[19]. Annotating cell identities is even more challenging for SRT datasets[20]. For example, spot-based protocols such as Visium[9], Slide-seq[13,14], HDST[15], and Stereo-seq[16] capture RNA from areas spanning more (or less) than one cell, without

[1]Tencent AI Lab, Shenzhen 518057, China. [2]BGI-Shenzhen, Shenzhen 518083, China. [3]Institute of Science and Technology for Brain-Inspired Intelligence, Fudan University, Shanghai 200433, China. [4]CAS Key Laboratory of Computational Biology, Bio-Med Big Data Center, Shanghai Institute of Nutrition and Health, University of Chinese Academy of Sciences, Chinese Academy of Sciences, Shanghai, China. [5]Guangdong Bigdata Engineering Technology Research Center for Life Sciences, Shenzhen 518083, China. [6]Guangdong Provincial Key Laboratory of Genome Read and Write, BGI-Shenzhen, Shenzhen 518120, China. [7]These authors contributed equally: Rongbo Shen, Lin Liu, Zihan Wu, Ying Zhang. ✉e-mail: liyuxiang@genomics.cn; xuxun@genomics.cn; jianhuayao@tencent.com

consideration of cell boundaries. In addition, due to inherited experimental design, the low transcript detection sensitivity of spot-based protocol further makes the measured transcriptomic profiles deviated from the real transcript levels of single cells. Also, FISH-based protocols face similar problem arising from potential inaccuracy of cell segmentation.

To completely understand the cell type organization of biological tissues, many cell atlas projects such as Human Cell Atlas (HCA)[21], Allen Brain Atlas (ABA)[22], Brain Atlas of BICCN[23], exploited hierarchical clustering scheme on large-scale scRNA-seq datasets to establish cell type taxonomy and define marker genes of each cell type by differential gene expression analysis. One of the main limitations of clustering methods is that the cell type taxonomy and marker gene sets are bound to a certain level of clustering resolution[24]. Moreover, the computational characterization of the cell heterogeneity requires a series of statistical per-cell gene signature assessments to remove low-quality and doublet-driven clusters[3,24]. Besides, facing the explosive data growth of new sequencing technologies, such de novo clustering methods[24] become more laborious, computationally intensive and inefficient. It is more desirable to develop supervised cell typing methods that transfer knowledge from annotated reference datasets to newly generated datasets[25].

To better characterize cell identities from reference datasets with well-defined cell types, many correlation-based and supervision-based cell typing methods have been introduced in scRNA-seq data analysis, such as SingleR[26], Scmap[27], Cell-ID[28], ScNym[29], SciBet[30], and ScDeepSort[31]. Specifically, SingleR[26] performed cell type annotation for newly generated dataset that was correlated to reference datasets of pure cell types and strengthened its inferences by reducing the reference datasets to only top cell types iteratively. Scmap[27] provided a cell typing strategy that projected newly generated dataset onto a reference dataset by searching the most similar clusters or cells (i.e., nearest neighbors) in the reference dataset and then assigning a cell type if its nearest neighbors have the same cell type. Cell-ID[28] independently extracted per-cell gene signatures for a newly generated dataset and reference datasets through multiple correspondence analysis, then used per-cell gene signatures to perform automatic cell type and functional annotation for target single-cell transcriptomic dataset by cell matching and label transferring from reference datasets. ScNym[29] employed an adversarial neural network to transfer cell identity annotations from a labeled reference dataset to a newly generated dataset despite biological and technical differences. SciBet[30] used the mean expression of cell type-specific genes selected by E-test to train a multinomial-distribution model, then calculated the likelihood function of a test cell using the trained model and annotated cell type for the test cell with maximum likelihood estimation. ScDeepSort[31] pretrained a weighted graph neural network (GNN)[32] to perform cell type annotation for newly generated datasets. However, applying these correlation-based and supervision-based cell typing methods to SRT datasets does not efficiently use the available spatial information which may be beneficial to cell type annotation.

By considering the characteristics of the SRT technologies, several integrated analysis methods[25,33] combined SRT data with scRNA-seq data to bridge these trade-offs and provide a better understanding of the spatial organization of tissues. The deconvolution-based methods[34–36] were a kind of primary integrated analysis methods, such as SPOTlight[34], RCTD[35], and Cell2location[36], that learned cell-type gene signatures from reference datasets to disentangle discrete cellular subpopulations from mixtures of mRNA transcripts from each capture spot in SRT datasets. Although the emerging SRT technologies, such as MERFISH, Slide-seq and Stereo-seq, had achieved cell-level or subcellular spatial resolution, some deconvolution-based methods, e.g., Cell2location, can still support cell-type annotation at cell-level spatial resolution. Specifically, Cell2location employed an interpretable hierarchical Bayesian model to map the spatial distribution of cell types by integrating scRNA-seq data and multi-cell SRT data from a same tissue. The mapping-based methods[37,38] were another kind of primary integrated analysis methods that mapped each scRNA-seq cell to a specific spot or cell of SRT data from a same tissue. Specifically, Seurat v3[37] provided a comprehensive integration strategy to map scRNA-seq data and SRT data, and projected cellular states (e.g., cell type) from a reference dataset to newly generated datasets. Tangram[38] employed a deep learning framework based on nonconvex optimization to align scRNA-seq data to various forms of spatial data collected from the same tissue, then mapped the cell types defined by scRNA-seq on the spatial context. However, these integrated analysis methods also did not efficiently use the available spatial information. Besides, several spatial domains analysis methods, such as SpaGCN[39], SEDR[40], STAGATE[41], were proposed to spatial domain clustering analysis that enable coherent gene expression in spatial domains by integrating gene expression and spatial location together. Inspired by the significant advancements of these spatial embedding-based clustering methods in identifying anatomical spatial domains, embedding spatial information should be beneficial to cell type annotation of SRT datasets.

In this study, we propose a cell typing method (SPATIAL cell type IDentification, Spatial-ID) that integrates transfer learning and spatial embedding strategies for high-throughput cell-level SRT datasets (Fig. 1). The transfer learning strategy employs scRNA-seq datasets with well-defined cell-type gene signatures collected from similar tissues to train deep neural network (DNN) models. Therefore, the cell type taxonomy of newly generated SRT datasets can also be aligned with existing cell atlas of scRNA-seq datasets that was constructed from similar tissues. To perform spatial embedding, we propose a graph convolution network (GCN)[32] that constructs a spatial neighbor graph by considering each cell as a node and the spatial location relationships between cells as edges. In the architecture of GCN, an autoencoder[42] is used to encode gene expression profiles and a variational graph autoencoder[43] is used to embed spatial information simultaneously. To handle the large number of cells in the high-throughput SRT data, we employ sparse convolution in GCN to accelerate the framework[44]. A self-supervised learning strategy is performed in GCN by constructing pseudo-labels from the probability distribution predicted by the DNN model of transfer learning[45].

The main contribution of the Spatial-ID is the effective incorporation of existing knowledge of reference scRNA-seq datasets and the spatial information of SRT datasets. A series of benchmarking analysis on publicly available SRT datasets with different data characteristics (see Supplementary Table 1) demonstrate the superiority of Spatial-ID in cell type annotation compared with other state-of-the-art methods (see Supplementary Table 2), i.e., Seurat v3, SingleR, Scmap, Cell-ID, ScNym, SciBet, Tangram, and Cell2location. Furthermore, Spatial-ID can effectively perform cell type annotation for 3D SRT datasets. Moreover, the extended process of new cell type discovery demonstrates that the predictions of Spatial-ID have a promising discrimination to detect new cell types. A group of simulation experiments with different gene dropout rates demonstrates more robustness of Spatial-ID than other state-of-the-art methods. A series of ablation analysis also proves the important of spatial information in Spatial-ID. These results suggest that embedding spatial information can substantially improve cell type annotation on SRT datasets. Besides, the application of Spatial-ID on a Stereo-seq SRT dataset with 3D spatial dimension shows its advancement on the large field tissues (~1cm²) with subcellular spatial resolution. The cell types identified by Spatial-ID present high consistency with previous studies. Besides, by mapping the identified cell types with identified spatial gene patterns, the significant GO (gene ontology) terms of the spatial gene patterns further reveal the functions and underlying biological processes of the identified cell types.

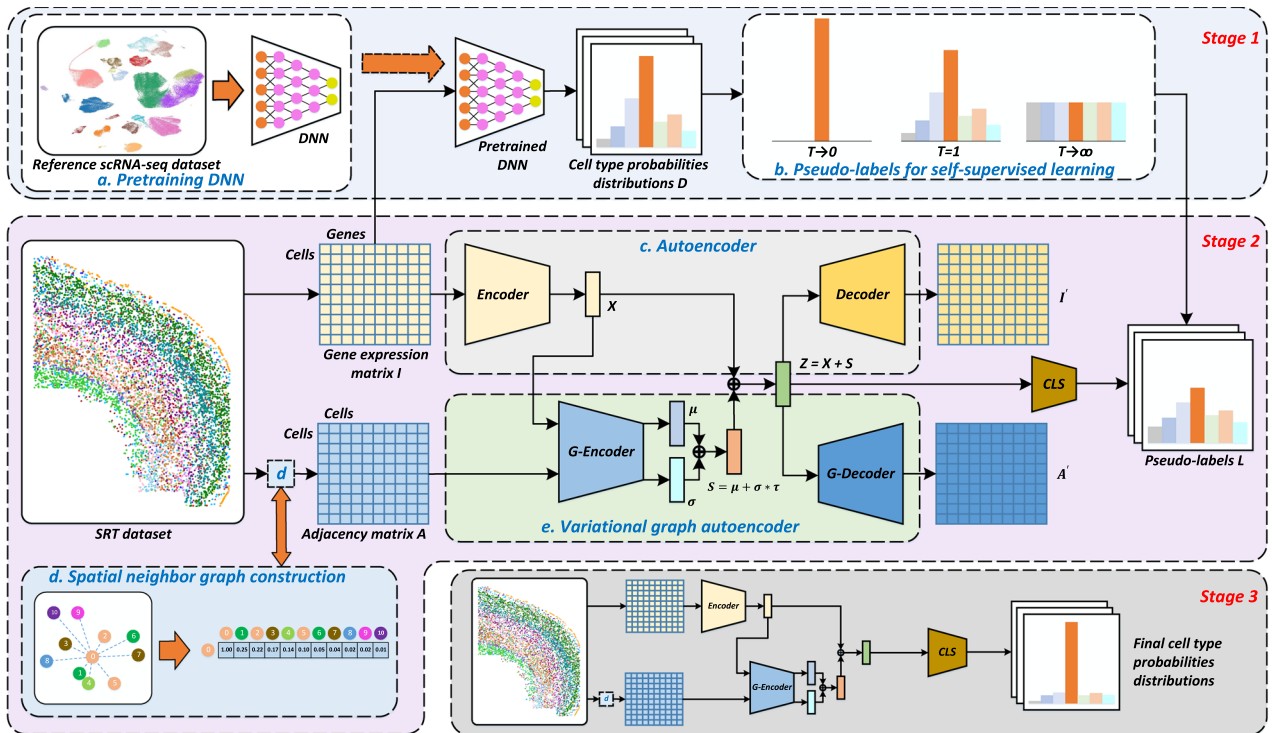

**Fig. 1 | Overview of Spatial-ID.** Stage 1 involves knowledge transfer from reference datasets. Stage 2 involves feature embedding of gene expression and spatial information, and employs self-supervised strategy to train a classifier (CLS) using the generated pseudo-labels in stage 1. Stage 3 uses the optimal model derived from Stage 2 to perform cell type annotation. **a** Reference scRNA-seq datasets are employed to pretrain deep neural network (DNN) models. **b** Based on the cell type probabilities distributions $D$ produced by pretrained DNN, pseudo-labels $L$ are generated by adjusting the temperature parameter $T$. **c** A deep autoencoder is used to learn encoded gene representation $X$ through reducing the dimension of the gene expression matrix $I$. The gene expression matrix $I'$ reconstructed by decoder is used to optimize the autoencoder by minimizing with the input gene expression matrix $I$. **d** A spatial neighbor graph is constructed to represent the spatial relationships between neighboring cells, where the relationship weight of each pair of cells is negatively associated with Euclidean distance. Therefore, the spatial neighbor graph is represented as an adjacency matrix $A$. **e** A variational graph autoencoder (VGAE, a kind of GCN) is used to embed the encoded gene representations $X$ from autoencoder and the adjacency matrix $A$, and then generate the spatial embedding $S$ as output. The reconstructed adjacency matrix $A'$ is used to optimize the VGAE by minimizing with the input adjacency matrix $A$.

## Results

### The pipeline of Spatial-ID

The pipeline of our proposed Spatial-ID is shown in Fig. 1, including 3 stages. Stage 1 involves knowledge transfer from reference datasets. Stage 2 involves feature embedding of gene expression profiles and spatial information of SRT dataset, and employs self-supervised strategy to train the classifier. Stage 3 uses the optimal model derived from stage 2 to perform cell type annotation for SRT dataset.

Formally, the DNN in stage 1 is trained on the gene expression profiles and ground truth labels of reference datasets (Fig. 1a). The pretrained DNN is used to generate the probability distribution for each cell of SRT dataset, then the probability distribution is used to construct pseudo-label (Fig. 1b) through a temperature setting strategy[45] (see "Methods"). The GCN in stage 2 contains an autoencoder, a variational graph autoencoder and a classifier. Given a SRT dataset, the gene expression profiles are transformed into a cell-gene matrix (i.e., gene expression matrix $I$), and a spatial neighbor graph (i.e., cell−cell adjacency matrix $A$) is constructed by considering each cell as a node and the spatial location relationships between cells as edges, where the relationship weight of each pair of cells is negatively associated with Euclidean distance (Fig. 1d). The autoencoder (Fig. 1c) takes the gene expression matrix and outputs the encoded gene representations $X$. The adjacency matrix $A$ and the encoded gene representations $X$ are fed into the variational graph autoencoder (Fig. 1e), that performs spatial embedding for the gene representations and outputs the final latent representations $Z$ that provide comprehensive characters of gene expression profiles and spatial information.

Thereafter, the final latent representations $Z$ are used to reconstruct the gene expression matrix in the autoencoder and the adjacency matrix in the variational graph autoencoder, respectively. Simultaneously, the self-supervised learning strategy employs the final latent representations $Z$ and the generated pseudo-labels $L$ from stage 1 to train the classifier. In stage 2, the training process optimizes the parameters of the GCN model until convergence, and saves the optimal model. Finally, the stage 3 (i.e., inference stage) reloads the optimal model from stage 2, and output the cell type predictions of a given SRT dataset.

To verify the advancement of the proposed Spatial-ID, we apply Spatial-ID on 5 representative SRT datasets with different characteristics (see Supplementary Table 1), including two publicly available mouse brain datasets (i.e., primary motor cortex[46] and hypothalamic preoptic region[47]) measured by MERFISH, a publicly available mouse spermatogenesis[48] dataset measured by Slide-seq, a publicly available human non-small-cell lung cancer[49] (NSCLC) dataset, and a self-collected mouse brain hemisphere dataset measured by Stereo-seq. The two FISH-based mouse brain datasets measured hundreds of genes, in which the mouse hypothalamic preoptic region dataset was a 3D SRT dataset that provided continuous slices with equal interval. The spot-based mouse spermatogenesis dataset and the self-collected mouse brain hemisphere dataset measured tens of thousands of genes, where the mouse brain hemisphere dataset provided 3D spatial dimension, subcellular spatial resolution and large field view on entire mouse brain hemisphere. The human NSCLC dataset measured 980 genes on formalin-fixed paraffin-embedded (FFPE) samples. We

employ the mouse primary motor cortex dataset, the mouse hypothalamic preoptic region, the mouse spermatogenesis dataset and the human NSCLC dataset to perform benchmarking analysis between Spatial-ID with other state-of-the-art methods (see Supplementary Table 2). The self-collected mouse brain hemisphere dataset is employed to demonstrate the advancement of Spatial-ID on spatial transcriptomic analysis of the large field tissues. To perform Spatial-ID and the control methods, we prepare 5 matched scRNA-seq datasets for the aforementioned SRT datasets (see Supplementary Table 1). Besides, to compare the robustness against gene dropout between Spatial-ID and the control methods, a simulation experiment of different gene dropout rate is performed on the FISH-based mouse primary motor cortex dataset. In addition, a postprocess of new cell type discovery is performed on the mouse primary motor cortex dataset and the human NSCLC dataset to demonstrate the discrimination of Spatial-ID.

## Application to mouse primary motor cortex dataset measured by MERFISH

We first perform a quantitative comparison between Spatial-ID and the control methods on the mouse primary motor cortex (MOP, see Fig. 2a) dataset[46] measured by MERFISH. The MOP dataset contains 12 samples, including total 280,186 cells and 254 genes. The snRNA-seq 10x v3 B dataset[50] (matched dataset of MOP dataset) is used as the trainset of the DNN model in Spatial-ID and the reference dataset in the control methods, which contains 159,738 cells and 31,053 genes. The MOP dataset and the snRNA-seq 10x v3 B dataset are derived from Brain Atlas of BICCN[23], thus the cell type assignments of them adapt the same MOP cell taxonomy that is a hierarchical organization with reference to the common cell type nomenclature (CCN)[51] found by Allen Institute. The cells are divided into excitatory neuronal cells (glutamatergic), inhibitory neuronal cells (GABAergic) and non-neuronal cell classes at the first level, then cells in each class are further divided into more subclasses based on their marker genes or spatial organization in cortex (Fig. 2b, d and Supplementary Fig. 1a). For example, the inhibitory neurons are further divided into five subclasses by marker genes: parvalbumin (Pvalb), somatostatin (Sst), vasoactive intestinal polypeptide (Vip), synuclein gamma (Sncg) and lysosomal-associated membrane protein family member 5 (Lamp5) (Supplementary Fig. 1c). The excitatory neurons are further divided into several layers with distinct projection properties (defined by known marker genes): intra-telencephalic neurons (L2/3 IT, L4/5 IT, L5 IT, L6 IT) (Supplementary Fig. 1b), extra-telencephalic projecting neurons (L5 ET), near-projecting neurons (L5/6 NP), corticothalamic projection neurons (L6 CT) and layer 6b neurons (L6b) (Fig. 2h). It should be noted that the number of cell types, the number of measured genes, relative abundance of cells in various cell types and the gene dropout rate differ in the snRNA-seq 10x v3 B dataset and the MOP dataset. For example, smooth muscle cells (SMC) and Sncg cells are depleted in the snRNA-seq 10x v3 B dataset, while Sncg cells are depleted in the MOP ST dataset. Besides, L6 IT Car3 and L4/5 IT are not provided in the snRNA-seq 10x v3 B dataset, thus these cell types are not considered in direct comparisons.

Compared with the control methods, Spatial-ID could effectively identify the cell types (Fig. 2c) and achieve better performance (Fig. 2f). On all 12 MOP samples, Spatial-ID achieves the highest mean accuracy 92.75% (Fig. 2f), and the differences with the control methods are very significant (Wilcoxon test $p$-value ≪ 0.001 for all other methods). Besides, Spatial-ID achieves mean weighted F1 score 0.9209 (Supplementary Fig. 2a), where weighted F1 score of each sample is calculated by weighted averaging the F1 score of each cell type, to mitigate the effects of cell type imbalance. If we abandon the spatial information, the DNN can achieves mean accuracy 91.96% on all the 12 samples (Supplementary Fig. 7d). Obviously, this ablation analysis demonstrates that it is beneficial to use spatial information in Spatial-ID. In

addition, Cell-ID achieves mean accuracy 17.08% and mean weighted F1 score 0.1521 that is far below other methods and is therefore not shown in Fig. 2f and Supplementary Fig. 2a. From this, each cell in MOP dataset have both a predicted cell type from the snRNA-seq 10x v3 B dataset and a ground truth label provided by MOP dataset. Cells in MOP dataset are grouped based on their ground truth labels, and then the fraction of predicted cell type for each group is determined (Fig. 2g and Supplementary Fig. 2d). Except the smooth muscle cells (SMC) and Sncg cells, Spatial-ID achieves excellent recall rates on other cell types. The depleted SMC and Sncg cell types in the snRNA-seq 10x v3 B dataset lead to weaker identification ability of Spatial-ID on these cell types, because there are no sufficient samples in training. Moreover, the integration of spatial information enables Spatial-ID to reveal that neuronal cells have specific spatial organization pattern (Fig. 2e). For the excitatory neurons, we can observe a laminar appearance for the overall cellular organization along the direction of the cortical depth, especially the IT neurons (Supplementary Fig. 1b). Obviously, the L2/3 IT, L5 IT, L6 IT neurons, identified by Spatial-ID, appear as discretely laminar cell populations, and L5 ET, L5/6 NP, L6 CT, and L6b neurons also appear as discrete cell populations (Fig. 2h). However, IT neurons and other types of excitatory neurons partially overlap in space, and more overlapping with inhibitory neurons and non-neurons lead to a high level of local cellular heterogeneity. To describe the spatial intermixing of different cell populations, we count the neighborhood cells of each cell to calculate the neighborhood complexity[47] and the neighborhood purity[47], and then estimate the neighborhood complexity distribution and neighborhood purity distribution for each identified cell type. Compared with the ground truth, the neighborhood purity distribution of Spatial-ID presents very similar characteristic (Jensen–Shannon distance: 0.013), and the neighborhood complexity distribution of Spatial-ID presents a slight shift to the lower complexity (Jensen–Shannon distance: 0.052) (Fig. 2i). Compared with the control methods (Supplementary Figs. 2b, 2c, 3 and 4), Spatial-ID also achieves high consistent of neighborhood complexity and neighborhood purity with ground truth.

Different SRT technologies usually have different rates of gene capture, especially the spot-based SRT technologies. To verify the robustness of Spatial-ID against different gene dropout rates, we conduct simulation experiments on MOP dataset by randomly discarding part of values in gene expression profiles (see "Methods"). Under the same configuration, Spatial-ID could achieve better performance of cell type annotation than the control methods (Fig. 2j). Especially in the low dropout rates (e.g., less than 0.6), the performance degradation of Spatial-ID is less than that of the top control methods (Fig. 2j and Supplementary Fig. 5a). Specifically, on all 12 MOP samples, Spatial-ID achieves the highest mean accuracy 85.76% at the dropout rate of 0.5 (Fig. 2j), and the differences with the control methods are very significant (Wilcoxon test $p$-value ≪ 0.001 for all other methods). While Spatial-ID achieves the highest mean weighted F1 score 0.8466 at the dropout rate of 0.5 (Supplementary Fig. 5b). These results suggest that the spatial information is not only beneficial to help cell type identification, but also improve the robustness of Spatial-ID against the variation of gene dropout (Fig. 2j, Supplementary Fig. 5c, d). Thus, Spatial-ID shows a promising perspective to transfer knowledge from available reference datasets (e.g., scRNA-seq datasets or other SRT datasets), even if their gene dropout rates differ from that of newly generated datasets.

L4/5 IT and L6 IT Car3 neurons are not annotated in the snRNA-seq 10x v3 B dataset, thus Spatial-ID predicts the L6 IT Car3 neurons as L6 IT neurons, and L4/5 IT neurons as L2/3 IT and L5 IT neurons (Fig. 2c, e). The ground truth of MOP dataset shows that L4/5 IT neurons populate in the continuous region between L2/3 IT and L5 IT neurons (Fig. 2d), and L6 IT Car3 neurons populate near the L6 IT neurons at L6 layer (Fig. 2d). These results suggest that the gene expression profiles of L4/5 IT neurons present gradual transition between the L2/3 IT and L5 IT

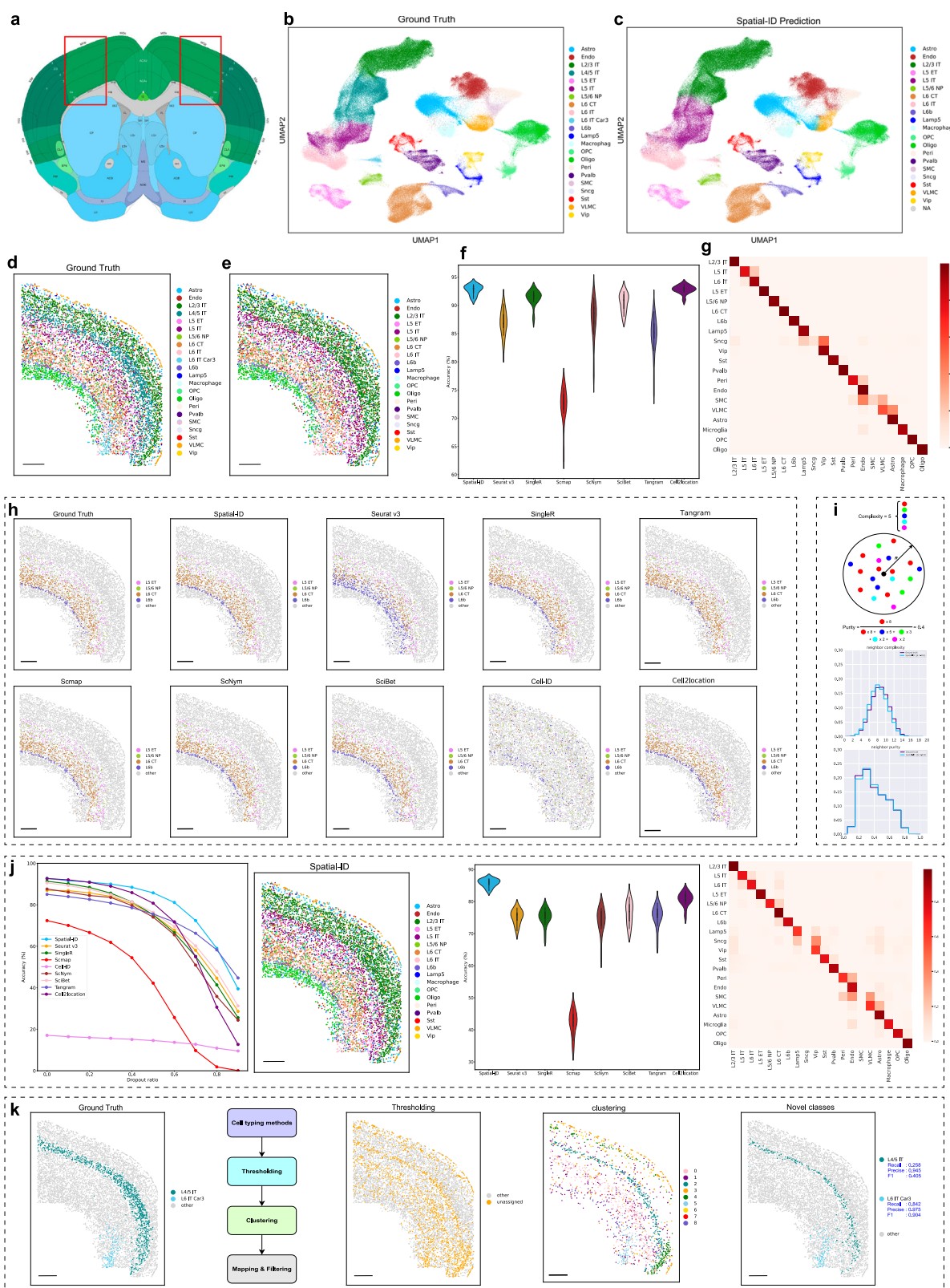

neurons, and the characteristics of L6 IT Car3 neurons are similar to the L6 IT neurons (Fig. 2c). We conduct a postprocess for Spatial-ID to further distinguish these new cell types that are presented in the MOP dataset but unseen in the snRNA-seq 10x v3 B dataset. Based on the predictions of Spatial-ID, a pipeline of new cell type discovery (Fig. 2k) is performed to determine whether there are new cell types in the MOP dataset, including thresholding, clustering, mapping, and filtering

(see "Methods"). The unassigned cells are determined by a threshold (i.e., 0.9) at the thresholding step. The clustering step is performed to group the unassigned cells into different clusters (i.e., 9 clusters). According to the clusters, the mapping step aligns the clusters of unassigned cells to predicted cell type in the same feature space, as shown in Supplementary Fig. 2g. Compared with the predictions of Spatial-ID (Supplementary Fig. 2f), each cluster contains several cell

**Fig. 2 | Application to mouse primary motor cortex dataset measured by MERFISH.** **a** The MOP region annotations in the Allen CCF v3 (http://atlas.brain-map.org/). **b** The ground truth cell types using UMAP embedding. **c** The Spatial-ID prediction using UMAP embedding. **d** Spatial organization of the ground truth cell types in a coronal slice (slice153). Bar scale 400 μm. **e** Spatial organization of the Spatial-ID prediction in **d**. Bar scale 400 μm. **f** The comparison of cell type annotation accuracy; $n = 12$ independent samples; center line, median; box limits, upper and lower quartiles; whiskers, 1.5× interquartile range. Notably, the mean accuracy of Cell-ID is 17.08%, that is far below those shown and is therefore not shown. **g** The confusion matrix of Spatial-ID prediction. The vertical axis and the horizonal axis list the ground truth cell types and the prediction of Spatial-ID, respectively. **h** The ground truth of L5 ET, L5/6 NP, L6 CT, and L6b neurons, and the prediction of Spatial-ID and the control methods. Bar scale 400 μm. **i** The neighborhood complexity of a given cell is defined as the number of different cell types presented

within a neighborhood of 100 μm in radius. The neighborhood purity of a given cell is defined as the fraction of the most abundant cell type to all cells in the neighborhood of 100 μm in radius. **j** Simulations of different gene dropout rates. From left to right, the comparison of cell type annotation accuracy at different gene dropout rates, spatial organization of the Spatial-ID prediction at the dropout rate of 0.5, the comparison of cell type annotation accuracy at the dropout rate of 0.5 ($n = 12$ independent samples; Center line, median; box limits, upper and lower quartiles; whiskers, 1.5× interquartile range), the confusion matrix of Spatial-ID prediction at the dropout rate of 0.5. Bar scale 400 μm. **k** New cell type discovery. From left to right, ground truth of L4/5 IT and L6 IT Car3 neurons, a pipeline of new cell type discovery, unassigned cells after thresholding, clusters derived from clustering for unassigned cells, and the finally found new cell types (i.e., L4/5 IT and L6 IT Car3). Bar scale 400um.

types predicted by Spatial-ID. For example, the cluster 2 contains the L2/3 IT and L5 IT cell types predicted by Spatial-ID, the cluster 5 contains the L6 IT cell type predicted by Spatial-ID. The filtering step analyzes the distance from the center of each cluster to the center of the predicted cell types. If the distance is less than the radius of one predicted cell type in feature space, the cluster is still identified as this predicted cell type rather than a new cell type. If the distance is larger than the radius of all predicted cell types in feature space, the cluster is identified as a new cell type. For example, the distance of cluster 2 to L2/3 IT cell types predicted by Spatial-ID in the feature space is larger than the radius of predicted L2/3 IT cell types, as well as L5 IT cell types predicted by Spatial-ID (Supplementary Figs. 2f, g). Therefore, the cluster 2 is identified as a new cell type that is labeled as L4/5 IT in ground truth (Supplementary Fig. 2e). Similarly, the cluster 5 is identified as a new cell type, that is labeled as L6 IT car3 in ground truth (Supplementary Fig. 2e). Applying the above filtering rule to the other clusters in turn, we can find that these clusters are closer to their main cell type predicted by Spatial-ID, therefore these clusters are not identified as new cell types. Finally, the discovered L4/5 IT and L6 IT Car3 neurons achieves F1 score of 0.405 and 0.904, respectively (Fig. 2k).

### Application to mouse hypothalamic preoptic region dataset measured by MERFISH

To quantitatively compare performance on 3D SRT dataset, we employ the mouse hypothalamic preoptic region (1.8 mm × 1.8 mm × 0.6 mm, Fig. 3a) dataset[47] measured by MERFISH to perform benchmarking analysis. This dataset measures a panel of 155 genes. We select 3 samples with naive behavior, including total 213,192 cells collected from 2 female mice and 1 male mouse (Fig. 3b). Each sample (Bregma 0.26 to −0.29) contains 12 slices with 50 μm interval. The reference scRNA-seq dataset (GSE113576)[47] is also collected from the hypothalamic preoptic region (~2.5 mm × 2.5 mm × 1.1 mm) across 3 replicates of an adult female mouse and a male mouse, including 31,299 cells and 27,998 genes. The delineation of major cell classes includes inhibitory neurons, excitatory neurons, microglia, astrocytes, immature oligodendrocytes, mature oligodendrocytes, ependymal cells, endothelial cells, macrophages, and mural cells.

Spatial-ID achieves the highest mean accuracy 87.74% (Fig. 3c, e) that significantly outperforms the control methods (Wilcoxon test p-value ≪ 0.001 for all other methods) on all 3 samples with naive behavior. The DNN can achieves mean accuracy 85.00% on all the 3 samples (Supplementary Fig. 7d). Obviously, spatial location information is also beneficial for Spatial-ID on the 3D SRT dataset. In the 3D views (Fig. 3d), We can also observe highly consistent cell type distribution of Spatial-ID with the ground truth, that shows obvious superiority than the control methods. Besides, Spatial-ID achieves the highest mean weighted F1 score 0.8773 (Supplementary Fig. 7a). These results suggest that Spatial-ID can be effectively applied on 3D SRT dataset. The cell types identified by Spatial-ID (Fig. 3f) show better

correspondence to ground truth than those identified by the control methods. Specifically, in the 2D views (Supplementary Fig. 6a, b), the annotation results of Spatial-ID also show better cell type distribution than the control methods. In addition, we introduce a set of experiments to compare the effect of spatial information (i.e., number of neighbor cells) on this dataset (Supplementary Fig. 7e). We can observe that the number of neighbor cells slightly affects the performance of accuracy, and the larger number of neighbor cells may lead to reduced differences between local cells.

### Application to mouse spermatogenesis dataset measured by Slide-seq

Next, we perform a benchmarking analysis on the mouse spermatogenesis dataset[48] (Fig. 4a) measured by Slide-seq. The mouse spermatogenesis dataset is acquired from three leptin-deficient diabetic (ob/ob) mice and three wild-type (WT) mice, including 207,335 cells in total and 24,105 genes in common. All cells are divided into 9 testicular cell types[52]: elongating/elongated spermatid (ES), round spermatid(RS), spermatocyte (SPC), spermatogonium (SPG), Endothelial, Sertoli, Leydig, Myoid and Macrophages (Fig. 4a). Testicular cell types are organized in a spatially segregated fashion at the level of seminiferous tubules. The reference scRNA-seq dataset (GSE112393)[53] includes 34,633 cells and 37,241 genes from an adult mouse testis. In order to perform Spatial-ID and the control methods, a shared gene set (24,105 genes) of the reference dataset is selected to align with the mouse spermatogenesis dataset.

The annotation result of Spatial-ID is shown in Fig. 4b. Based on the quantitative comparison, Spatial-ID also demonstrates superior accuracy of cell type identification on the mouse spermatogenesis dataset (Fig. 4c). Specifically, Spatial-ID achieves mean accuracy 60.45% on all the 6 samples (Fig. 4c) and achieves mean weighted F1 score 0.55 (Supplementary Fig. 7b). The ablation analysis shows that the DNN achieves mean accuracy 58.27% on all the 6 samples (Supplementary Fig. 7d). This again illustrates the importance of spatial information. Notably, Cell2location presents excellent performance and achieves the best accuracy 62.88% on mouse spermatogenesis dataset. The Slide-seq technology provides an approximate cell-level spatial resolution, where each spot contains several cells (i.e., 1 to 10). Technically, this dataset is more suitable for deconvolution-based methods. Besides, the ground truth of this dataset is derived from a non-negative matrix factorization regression (NMF) method (see Supplementary Table 1), that may introduce method bias for ground truth labels. This may allow regression-based methods, such as Cell2location, to achieve better results. Nevertheless, Spatial-ID achieves comparable accuracy with Cell2location. These results suggest that Spatial-ID can also effectively handle the spot-based Slide-seq dataset with tens of thousands of genes, even if the area of a spot spans more than one cell. As analysis in related study[48], diabetic induces testicular injuries through disrupting the spatial structures of seminiferous tubules and changing the expression pattern of many genes at

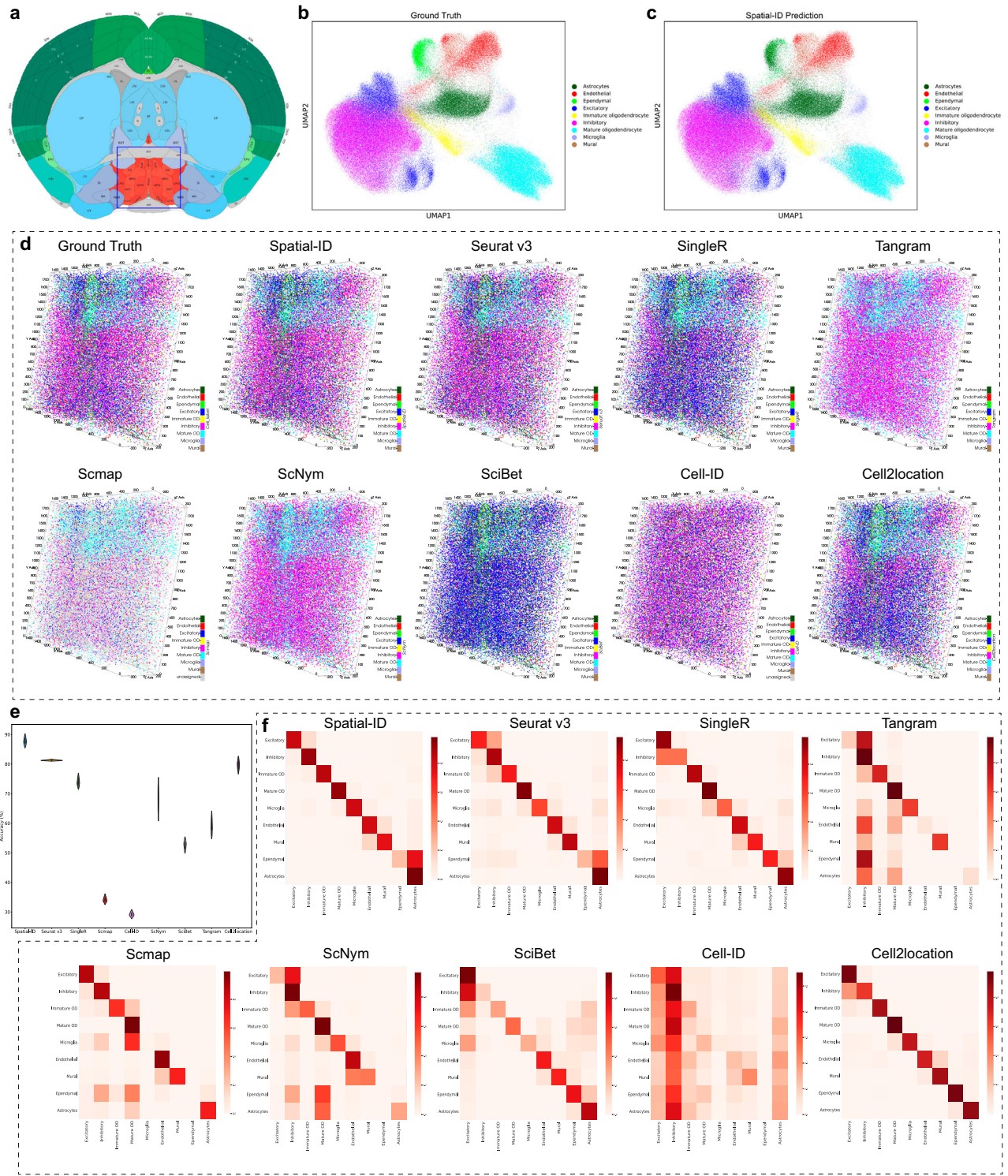

**Fig. 3 | Application to mouse hypothalamic preoptic region dataset measured by MERFISH. a** The mouse hypothalamic preoptic region annotations in the Allen CCF v3 (http://atlas.brain-map.org/). **b** Visualization of the ground truth cell types using UMAP embedding. **c** Visualization of the Spatial-ID predictions using UMAP embedding. **d** 3D spatial organization of the ground truth cell types of a sample with naive behavior, and the predictions of Spatial-ID and the control methods. Scale unit (μm). **e** The comparison of cell type annotation accuracy; *n* = 3 independent samples; Center line, median; box limits, upper and lower quartiles; whiskers, 1.5× inter-quartile range. **f** The confusion matrixes show the fraction of cells from any ground truth cell type predicted by Spatial-ID and control methods. The vertical axis lists the ground truth cell types, and the horizonal axis lists the predicted cell types. Mature OD: Mature oligodendrocyte; Immature OD: Immature oligodendrocyte.

molecular level. We can observe relatively regular spatial structure of seminiferous tubules in wild-type mouse (Fig. 4d), and irregular spatial structure of seminiferous tubules in diabetic mouse (Fig. 4e).

Moreover, we compare the runtime of Spatial-ID and the control methods on this SRT dataset (Fig. 4f). The running efficiency of Spatial-ID, ScNym and SciBet is much higher than other methods. The same results can be obtained for other SRT datasets (see Supplementary Table 3). Notably, although Tangram and Cell2location all employ GPU acceleration, they present lower running efficiency than Spatial-ID. To further analyze the running efficiency of Spatial-ID on different

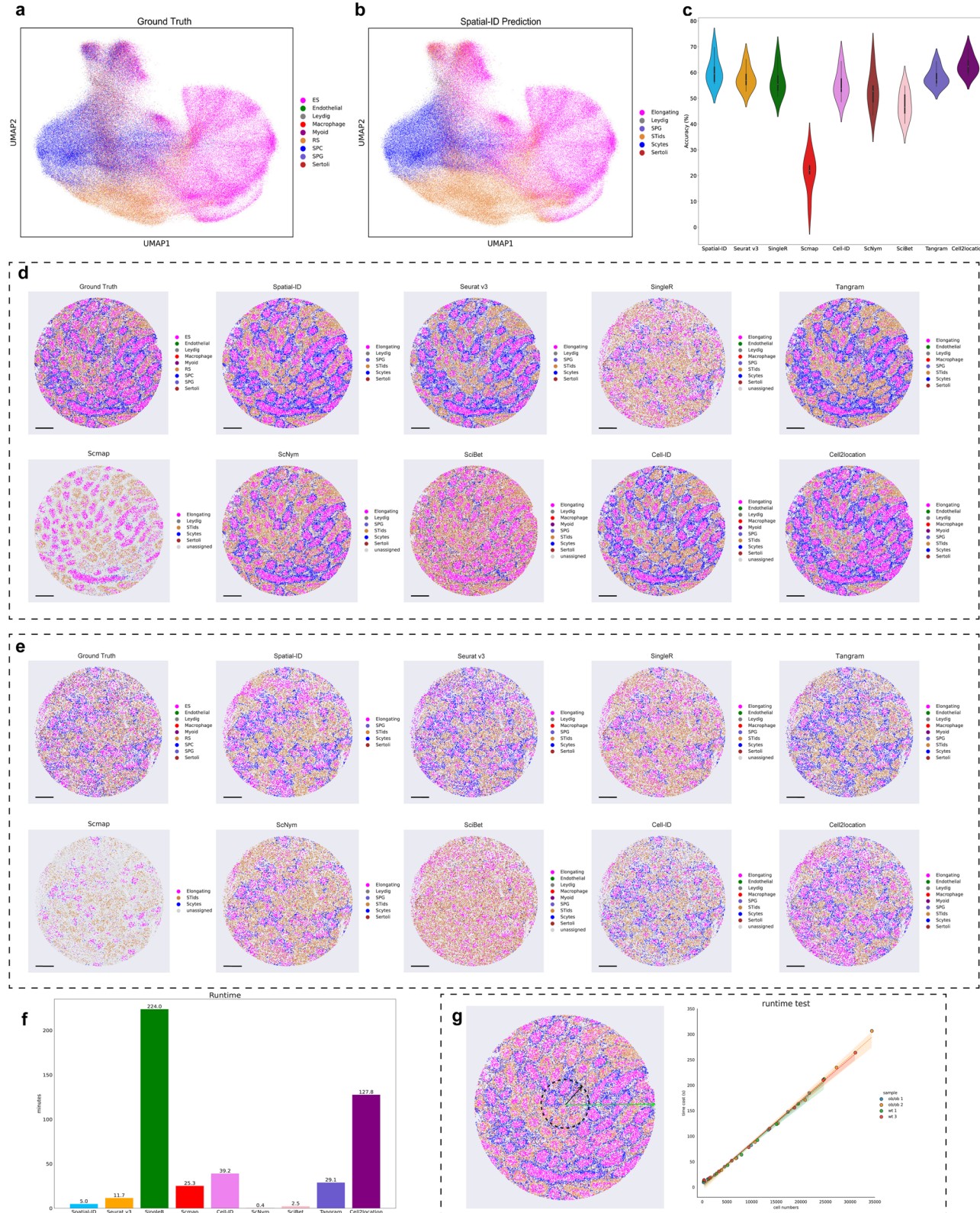

**Fig. 4 | Application to mouse spermatogenesis dataset measured by Slide-seq.** **a** Visualization of the ground truth cell types using UMAP embedding. ES elongating spermatid, RS round spermatid, SPC spermatocyte, SPG spermatogonium. **b** Visualization of the Spatial-ID predictions using UMAP embedding. **c** The comparison of cell type annotation accuracy; *n* = 6 independent samples; Center line, median; box limits, upper and lower quartiles; whiskers, 1.5× interquartile range. **d** Spatial organization of the ground truth cell types of a wild-type sample, and the predictions of Spatial-ID and the control methods. Bar scale 400 μm. **e** Spatial organization of the ground truth cell types of an ob/ob sample, and the predictions of Spatial-ID and the control methods. Bar scale 400 μm. **f** The average time cost per sample of Spatial-ID and control methods in this mouse spermatogenesis dataset. The comprehensive results for all SRT datasets in this study can be found in Supplementary Table 3. **g** The running efficiency analysis. The left one shows the scheme of field view sampling. The right one shows that the runtime of Spatial-ID increases linearly as the number of cells increases. The regression plots of runtimes are presented as mean values with 95% confidence intervals.

number of cells, we crop 10 percent to 100 percent radius field views of 4 samples (other 2 samples have non-circular field views) of this SRT dataset (Fig. 4g), respectively. Then, we statistically analyze the running efficiency of Spatial-ID on these cropped field views. As the number of cells increases, the runtime of Spatial-ID increases linearly (Fig. 4g).

## Application to human NSCLC dataset

We also perform a benchmarking analysis on a human non-small-cell lung cancer (NSCLC)[49] SRT dataset. We select the individual dataset Lung 9-1, including 20 samples and total 83,621 cells. These samples are generated from a Formalin-Fixed Paraffin-Embedded (FFPE) sample of a 60+ years old patient by high-plex spatial molecular imaging (0.18 μm per pixel) · CosMx SMI platform (www.nanostring.com). A panel of 980 genes are measured, and the field size of each sample is about 0.7 mm × 0.9 mm. The reference scRNA-seq dataset (scRNA-seq NSCLC)[54] includes 49,532 cells and 22,180 genes. The shared cell types contain mDC, Treg, fibroblast, T CD8, plasmablast, mast, T CD4, neutrophil, NK, macrophage, epithelial, pDC, endothelial, B-cell, and tumors.

Figure 5a shows the accuracy performance of Spatial-ID and the control methods, where Spatial-ID achieves the highest mean accuracy 69.76% on all 20 samples. Besides, Spatial-ID achieves mean weighted F1 score 0.6288 (Supplementary Fig. 7c). The ablation analysis shows that the DNN achieves mean accuracy 68.09% on all 20 samples (Supplementary Fig. 7d). The number of cells in some cell types of the human NSCLC dataset are scarce (Fig. 5b), and the ground truth labels of cells present indistinguishable communities in the feature space. We can observe that tumor cells predominate in this dataset (Fig. 5b, c). Affected by these factors, Spatial-ID misses some rare cell types, even though spatial-ID achieves the highest accuracy 87.73% on the example shown in Fig. 5d. To alleviate this problem, we employ the pipeline of new cell type discovery for the predictions of Spatial-ID (Fig. 5e). The thresholding step determines a set of unassigned cells by a threshold 0.7 (Fig. 5f). Then, these unassigned cells are grouped into 5 clusters at the clustering step (Fig. 5g). Next, the mapping step aligns the clusters to predicted cell types in the feature space for these unassigned cells (Fig. 5g, e). Finally, the filtering step analyzes the distance from the center of each cluster to the center of the predicted cell types. The cluster 0 and cluster 4 are identified as new cell types (Fig. 5h). Obviously, there are labeled as neutrophil cells and endothelial cells in ground truth. Although the postprocess of new cell type discovery can retrieve 2 missed cell types, there are still other missed cell types that are more scarce and indistinguishable. To further identify these cell types, more genes may need to be measured.

## Application to large field mouse brain hemisphere dataset measured by Stereo-seq

Many currently available sequencing-based SRT technologies such as Slide-seq, DBiT-seq[55], and HDST do not have single-cell spatial resolution, where each spot contains 1 to 10 cells. With the continued improvement of spatial resolution, newly emerging SRT technologies such as Seq-Scope[56] and Stereo-seq can produce high-throughput subcellular SRT data with a large number of cells in large field tissues in subcellular spatial resolution. Here, we generate single-cell spatial gene expression profiles of 3 adjacent coronal sections (10-μm thick, without intervals) along the anterior–posterior axis (Bregma −3.56 to −3.66) of right mouse brain hemisphere (Fig. 6a) using Stereo-seq. After the standard data processing and quality control (see "Methods"), 140,816 cells are retained for these 3 sections. The single-cell mouse brain atlas of cell types from the Linnarsson Lab[57] is employed as the reference dataset. After selecting the cell types of reference dataset that located in brain sections of our samples, a subset of 113,488 cells belonging to 152 cell types with a total of 747 marker genes is used as training set of the proposed Spatial-ID.

Based on the predictions of Spatial-ID, the identified cell types of 3 coronal sections present a high consistency (Fig. 6b) that an average of 99% cells in each section are assigned to their common cell types (Supplementary Fig. 8e). Besides, by color-coding our identified cell types in low dimension feature space (i.e., UMAP), most identified cell types congregate into distinguishable communities (Fig. 6b). For example, DGGRC2, VLMC2, and MBDOP2 can be easily segregated because they are driven by large differences of gene expression. Some similar cell types populate in mixed communities, such as the cortical pyramidal neuron (TEGLUs). According to the cell type taxonomy of reference dataset, 65,174 cells (50.8%) are identified as excitatory neurons, 20,267 cells (15.8%) are identified as inhibitory neurons, and 42,840 cells (33.4%) are identified as non-neuronal cell types (Fig. 6c). Specifically, most of the identified excitatory neurons are telencephalon projecting neurons with glutamatergic neurotransmitter (TEGLU) and populate in cerebral cortex and hippocampus, such as TEGLU4, TEGLU7 and TEGLU8 in cerebral cortex (Fig. 6e, j), TEGLU24 (Supplementary Fig. 9a) and DGGRC2 (Fig. 6k) in hippocampus. The other excitatory neurons with glutamatergic neurotransmitter (MEGLU, HBGLU) populate in midbrain and hindbrain. The identified inhibitory neurons mainly consist of TEINH19 and MEINH8 (Fig. 6i), where the identified TEINH19 neurons scatter across cortical layers and hippocampus CA3 region and the identified MEINH8 populate in midbrain. The identified non-neuronal cells exhibit a dispersed distribution throughout the mouse brain hemisphere, such as ACNT2 (non-telencephalon astrocyte cells) in midbrain, hindbrain, and fiber tracts (Supplementary Fig. 9a), VLMC2 (vascular leptomeningeal cells) at the interface of the brain structure (Supplementary Fig. 9a).

As the visualization of cell type annotation in Fig. 6b, most of identified cells show a distinguishable spatial distribution. To further reveal the anatomical functions of identified cell types throughout distinct brain regions[58–60], the entire right mouse brain hemisphere can be roughly split into several spatial anatomical functional regions according to the Allen Brain Atlas[61] (ABA; https://atlas.brain-map.org/), including isocortex, hippocampal formation, olfactory area, midbrain, hindbrain, interbrain, fiber tracts and vascular system. By quantifying the cells among these 8 regions, we can observe that different functional regions have different combinations of the identified cell types (Fig. 6d). For example, in the isocortex region (Fig. 6e), the identified cortical pyramidal neurons (TEGLU7, TEGLU8, TEGLU10, TEGLU4, TEGLU3, TEGLU2, etc.) display a layered laminar appearance along the direction of the cortical depth[62–65]. Moreover, we further illustrate a continuous gradient of cells along the cortical depth from L2/3 to L6 in the VISp and AUD regions (Fig. 6e). The cell type compositions of the VISp and AUD regions have significant differences, where fewer TEGLU3 and TEGLU8 populate in the AUD region than the VISp region (Fig. 6e).

For the mainly identified cell types, we further analyze the gene expression specificity of typical marker genes provided by the reference dataset[57] (Fig. 6f). Most of these marker genes have the highest expression in their corresponding cell types that have a relatively high fraction, e.g., Lamp5 of TEGLU7, Spink8 of TEGLU24, Slc6a3 of MBDOP2, Irx2 of HBGLU7, Carpt of MEGLU14, Opalin of MOL1, Mgp of ABC (Fig. 6g), etc. Moreover, several marker genes, such as Sv2b of TEGLU21, Lingo1 of TEGLU11 and Sst of TEINH19, present continuous expressions across the identified neurons in cerebral cortex and hippocampus (Fig. 6g). Interestingly, we observe the ACNT2 maker gene Slc6a11 expressed higher in ACNT1, another subclass of non-telencephalon astrocytes, than in ACNT2 (Fig. 6g). These observations may be derived from the continuous variation among neighborhood subclasses or the combinatorial expression of marker genes.

We further investigate the spatially varying genes indicative of the underling cell types, and their grouped spatial patterns[66]. Based on the notion that genes expressed at similar levels by proximal cells must be varying in an informative manner, Hotspot[67] allows the identification

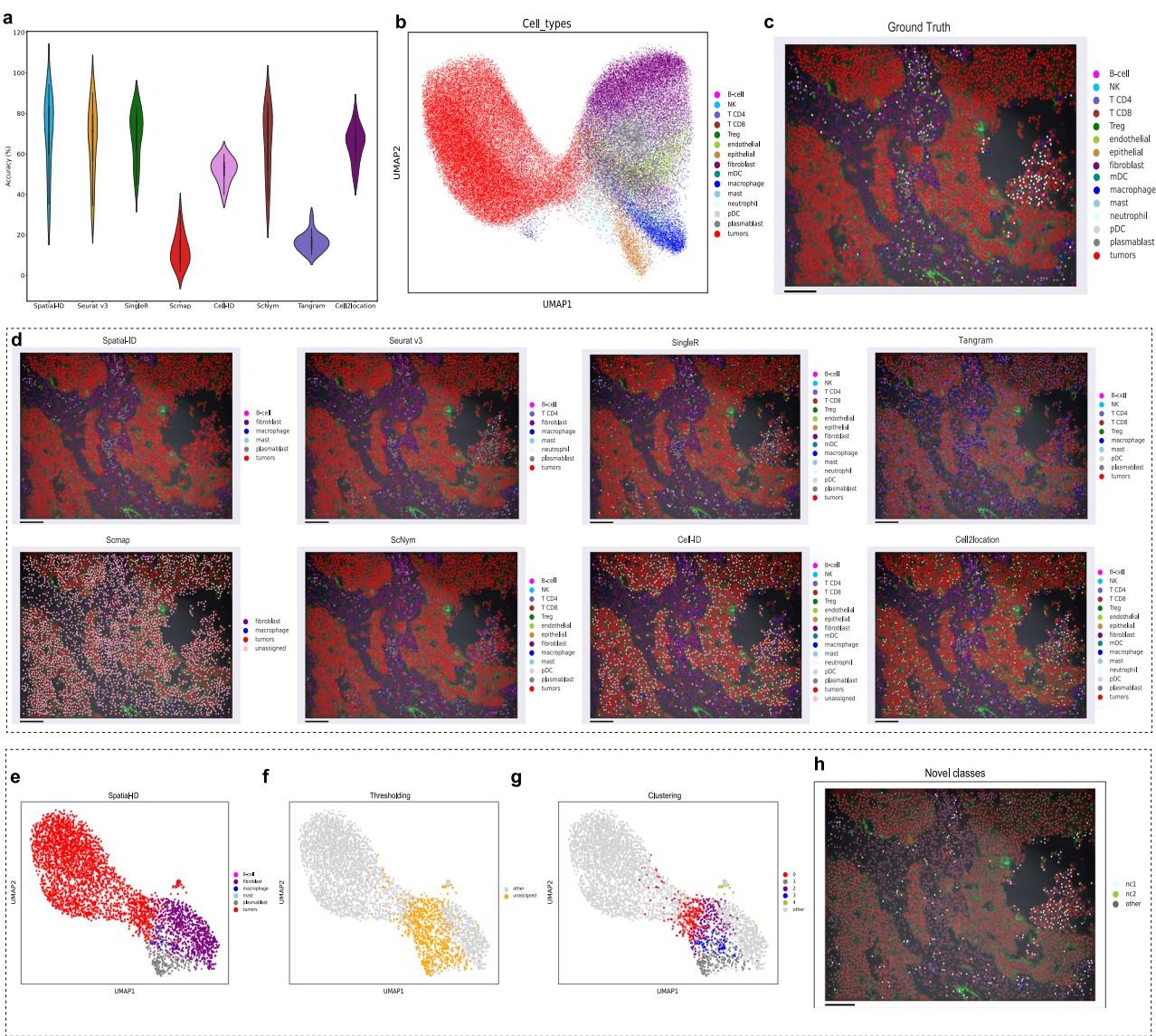

**Fig. 5 | Application to human NSCLC dataset. a** The comparison of cell type annotation accuracy; $n = 20$ independent samples; Center line, median; box limits, upper and lower quartiles; whiskers, 1.5× interquartile range. Notably, the mean accuracy of SciBet is 0.98%, that is far below those shown and is therefore not shown. **b** Visualization of the ground truth cell types using UMAP embedding. **c** Spatial organization of the ground truth cell types of a sample. Field size is about 0.7 mm × 0.9 mm. Bar scale 100 μm. **d** Spatial organization of the predictions of Spatial-ID and the control methods. Bar scale 100 μm. **e** Visualization of the predictions of Spatial-ID for this sample using UMAP embedding. **f** Visualization of the unassigned cells of this sample using UMAP embedding. **g** Visualization of the clusters of the unassigned cells using UMAP embedding. **h** Spatial organization of the finally found new cell types. nc1: new class type 1; nc2: new class type 2. Bar scale 100 μm.

of spatially informative genes and spatially dependent patterns of gene expression in situ for SRT datasets. A total of 30 specific spatial gene patterns[66] are detected by Hotspot (Supplementary Fig. 9b) and 6 of them are illustrated in Fig. 6h, i, j, and k. Notably, a spatial gene pattern may consist of type-specific genes from an individual identified cell type (Fig. 6h), but it may also be constituted by region-specific genes from diverse identified cell types (Fig. 6i, j, k). Specifically, the spatial gene pattern P26 is detected in the retrosplenial area of layer 2 (Fig. 6h), which includes Tshz2 (Supplementary Fig. 9d), one of the marker genes of the cortical projection neurons TEGLU6 that are identified in this region (Fig. 6h). The GO-based enrichment result indicates that the spatial gene pattern P26 may involve in myelination and axon ensheathment of central nervous system, possibly supporting a role for retrosplenial cortex in spatial coding, memory formation, and information integration[68,69] (Fig. 6l). In the midbrain, the obviously spatial gene pattern P17 (Fig. 6h), contains gene Ucn, Slc5a7, Chodl, etc (Supplementary Fig. 9d), significantly enriches in the sub-region of

dorsal raphe nucleus (DRN)[70]. Accordingly, the identified MEGLU14 neurons (marked genes: Cartpt, Ucn, and Chodl) specifically populate in this region (Fig. 6h). DRN has been implicated in disorder of anxiety, reward processing, as well as social isolation[71,72]. Here, we find that these DRN-specific genes are highly enriched at axon terminus, neuron projection terminus and terminal bouton (Fig. 6l), which are specialized to release neurotransmitters to transmit impulses between neurons. Another spatial gene pattern P10 is detected in the sub-regions of ventral tegmental area and subtantia nigra (SNr, Fig. 6h), contains genes Slc6a3, Slc18a2 and Th, etc (Supplementary Fig. 9d). This spatial gene pattern corresponds to identified MBDOP2 neurons (marked genes: Slc6a3 and Chrna6), which are dopaminergic neurons in midbrain that have been reported to be associated with the genetic risk of neuropsychiatric disorders[73], for example Parkinson's disease. The further GO-based enrichment result indicates that these enriched genes may involve in the regulation of neurotransmitter levels (Fig. 6l), revealing the relationship between gene expression of MBDOP2

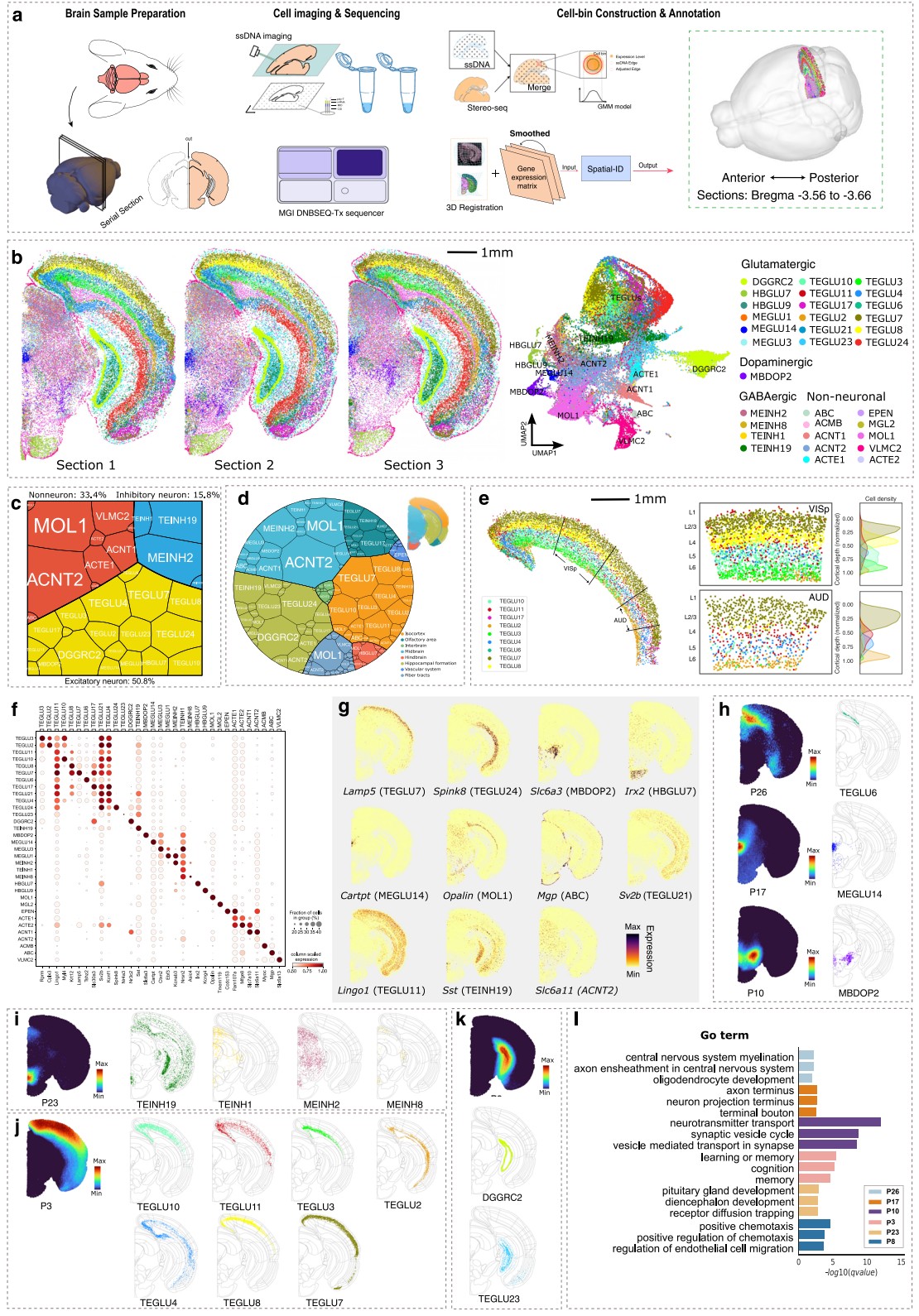

neurons and neuropsychiatric disorders again. Besides, serval identified spatial gene patterns do not indicate to specific cell types, such as P23 (Fig. 6i), P3 (Fig. 6j), and P8 (Fig. 6k). Particularly, P23, which is highly concentrated in the interpeduncular nucleus (IPN) of ventral midbrain, consists of several GABAergic neuron-associated genes (e.g., Otp, Pax7, Gad1, Gad2, Slc32a), suggesting its representative role for inhibitory neurons. As expected, we can observe that the identified

inhibitory neurons including TEINH19, MEINH2, MEINH8, and TEINH1 populate in this area (Fig. 6i). Previous studies found that IPN was the critical brain area associated with the reinforcing effects of nicotine[74]. The GO-based enrichment result reveals these spatially enriched genes are highly related with pituitary gland and diencephalon development and receptor diffusion trapping (Fig. 6l). P3 is highly concentrated in the isocortex (Fig. 6j), which indicates the identified cortical pyramidal

**Fig. 6 | Application to large field mouse brain hemisphere dataset measured by Stereo-seq. a** The workflow of data acquisition, data processing, and cell type annotation. **b** Cell type annotation of Spatial-ID for the 3 adjacent sections (Bregma −3.56 to −3.66 mm), and UMAP visualization. **c** A Voronoi treemap shows the composition of excitatory neurons, inhibitory neurons, and non-neuronal cells among the 3 sections. Every tile denotes one cell type and its size represents cell number. **d** A Voronoi diagram shows cell type organization among distinct brain regions of the 3 sections. Every tile is colored by its populated ABA functional region and its size represents cell number. **e** Spatial organization of the cortical pyramidal neurons, i.e., TEGLU2, TEGLU3, TEGLU4, TEGLU6, TEGLU7, TEGLU8, TEGLU10, TEGL11, and TEGLU17 in the Section 3. Cells in the VISp and AUD region are individually presented in the middle panel. The right panel shows the kernel

density estimate plots for the corresponding cell types along the normalized cortical depth. **f** The expression dot plots show the gene expression specificity of typical marker genes for identified cell types. Dot size represents the proportion of expressing cells and color indicates average expression level in each identified cell type. **g** Spatial distributions of selected marker genes show the number of transcripts captured by Stereo-seq. **h** The spatial gene patterns consist of type-specific genes (Section 3, visualized with pattern scores). The right panel shows the corresponding identified cell types together with the ABA spatial anatomical functional regions. **i–k** The spatial gene patterns consist of region-specific genes from diverse identified cell types (Section 3). The corresponding identified cell types are illustrated on the right. **l** Top three highly enriched GO terms for each spatial gene pattern in (**h–k**).

neurons (TEGLUs shown in Fig. 6e), and significantly enriched in GO terms of learning, memory and cognition (Fig. 6l). P8 is found to be involved in positive chemotaxis, and mainly constituted by genes from hippocampal neurons DGGRC2 and TEGLU23 (Fig. 6k).

## Discussion

In this work, we propose Spatial-ID to perform cell type annotation for SRT datasets molecularly. We first conduct a series of comparisons on 4 publicly available SRT datasets with different characteristics (see Supplementary Table 1). By comparing the accuracy and weighted F1 score calculated from predictions and ground truth, the proposed Spatial-ID demonstrates superior performance than the state-of-the-art methods (see Supplementary Table 2) on the benchmarking analysis. Compared with the results of DNN, this ablation analysis also shows that the spatial information plays an important role in Spatial-ID. Furthermore, the better performance of Spatial-ID on the 3D FISH-based SRT dataset (i.e., mouse hypothalamic preoptic region) further confirms that Spatial-ID can be effectively applied to 3D SRT dataset. Moreover, the comparisons on the randomly gene dropout simulations of the FISH-based SRT dataset (i.e., mouse primary motor cortex) additionally demonstrate the better robustness of Spatial-ID against the variation of gene dropout. These results suggest that embedding spatial information can not only improve the accuracy of cell type identification substantially, but also improve the robustness against the variation of sequencing technologies, such as different gene dropout rates, different number of measured genes. Besides, the extended process of new cell type discovery enhances the adaptability of Spatial-ID in the situation of transfer learning from an incomplete reference dataset or applications to the SRT datasets with scarce and indistinguishable cell types. In addition, the running efficiency of Spatial-ID on all SRT datasets (see Supplementary Table 3) is much faster than that of correlation-based methods (i.e., Seurat v3, SingleR, Scmap, and Cell-ID) and integrated methods (i.e., Tangram and Cell2location with GPU acceleration).

As the application of Spatial-ID on the large field mouse brain hemisphere dataset measured by Stereo-seq technology, we investigate the spatial organization of identified cell types that show high consistency with previous studies at spatial anatomical structure level, such as Allen Brain Atlas. By analyzing the gene expression specificity of typical marker genes reported in reference dataset, i.e., the single-cell mouse brain atlas of cell types from the Linnarsson Lab[57], most of these pre-defined marker genes have the highest expression in their corresponding cell types. Besides, by mapping the identified cell types with identified spatial gene patterns, the significant GO terms of the spatial gene patterns further reveal the functions and underlying biological processes of the identified cell types in mouse brain. Therefore, Spatial-ID shows a promising perspective to build a large-field spatial transcriptomic brain atlas.

In principle, the proposed Spatial-ID is a reference-based supervised cell typing method, therefore Spatial-ID is influenced by the characteristics of the reference datasets such as cell number, cell heterogeneity, the gene set used as features. For example, the depleted

cell types in reference dataset lead to insufficient knowledge transfer of Spatial-ID, that may present weaker identification ability on these cell types. Besides, the Spatial-ID is influenced by the hyperparameters of constructing graph network such as the number of nearest cells and coefficient of inverse distance weight (see Supplementary Table 4), that also indicates the intensity of spatial information used. Using more spatial information (neighbor cells) may lead to reduced differences between local cells, especially the scarce cell types (Figs. 4d, e and 5d). Therefore, we anticipate that the more spatial information used enables the Spatial-ID to enhance the identification of locally enriched cell types and inhibit the identification of rare cell type. In addition, the Spatial-ID demonstrates well robustness against the variation of gene dropout, but it is inevitably influenced by the high dimensionality and biological noise associated with SRT sequencing technologies. Beside optimizing the data preprocessing, integrating the SRT datasets with scRNA-seq datasets, that have been determined with rich mRNA transcripts, or other modalities, such as microscopy data and proteomics data, is potential to mitigate the sequencing noise.

## Methods
### Ethics declarations
The Stereo-seq dataset in this study was collected from a 5-week-old C57BL/6J laboratory male mouse. All experimental protocols for generating Stereo-seq dataset presented in this study were compliant with ethical regulations regarding animal research and approved by the Animal Care and Use committee of the Guangzhou Institutes of Biomedicine and Health, Chinese Academy of Sciences and the Institutional Review Board of BGI, Shenzhen, China (Ethical permit license number BGI-IRB A21001). All efforts were made to utilize only the minimum number of animals necessary to produce reliable scientific data.

### Overview of datasets
All experimental protocols for generating Stereo-seq dataset presented in this study were compliant with ethical regulations regarding animal research and approved by the Animal Care and Use committee of the Guangzhou Institutes of Biomedicine and Health, Chinese Academy of Sciences and the Institutional Review Board of BGI, Shenzhen, China (Ethical permit license number: BGI-IRB A21001). Our proposed Spatial-ID was performed on different SRT datasets. For each group of applications, we also collected reference scRNA-seq datasets. All other SRT datasets and reference scRNA-seq datasets are publicly available, as shown in Supplementary Table 1. Notably, the mouse hypothalamic preoptic region dataset measured by MERFISH and the self-collected mouse brain hemisphere measured by Stereo-seq are 3D SRT datasets. In addition, the field size of the self-collected mouse brain hemisphere is much larger than other public datasets.

All the ground truth labels we used were provided with the data of the original research papers (see Supplementary Table 1). In these papers, these cell type labels were annotated by combining computational analysis (e.g., clustering analysis or matrix factorization based analysis) and manual labeling of biologist and histologist. It is the

involvement of manual labeling that incorporated the established knowledge and guided the computational analysis, making the given label reliable. For example, the mouse primary motor cortex dataset that involves clustering analysis and manually filtering low-quality or doublet-driven clusters, shows more distinguishable characteristic in the feature space. Such practice is the most widely accepted standard in cell type annotation. The established knowledge includes known marker genes of specific cell types and known tissue structures inferred by the spatial context.

## Data collection of large field mouse brain hemisphere

The large field mouse brain hemisphere dataset was measured by Stereo-seq[16] (Fig. 6a). Briefly, three adjacent coronal sections (10-μm thick, without intervals) along the anterior–posterior axis (Bregma −3.56 to −3.66) were cut from the brain of a 5-week-old C57BL/6J male mouse (Fig. 6a). The Mouse was housed under standard laboratory conditions (12 h light/12 h dark cycle, temperature of 21–27 °C, and humidity of 55–60%) with ad libitum access to water and mouse chow. First, these sections were adhered to the DNA nanoball (DNB, i.e., spot) patterned Stereo-seq chip surface, and then were stained and scanned into ssDNA images for cellular localization. Each DNB on Stereo-seq chip contains a 25 bp randomly barcoded sequence as coordinate identity (CID) for its unique spatial location, a 10 bp molecular identifier (MID), and a polyT for in situ RNA capture, having a size of 220 nm in diameter and 500 nm center-to-center distance. Next, tissue permeabilization, in situ reverse transcription, amplification, library construction, and sequencing were performed according to the protocol of Stereo-seq technology. More specific process can be found in the Supplementary information of Stereo-seq technique.

## Data processing of large field mouse brain hemisphere

CID sequences were first parsed to the spatial coordinates of the DNB patterned silicon chip, allowing 1 base mismatch to correct for sequencing and PCR errors. Qualified reads (Q score ≥ 10) were then aligned to the mouse reference genome Ensembl GRCm38 v86 via STAR. Mapped reads with MAPQ ≥ 10 were annotated, and counted by handleBam (available at https://github.com/BGIResearch/handleBam). A resulting CID-containing expression profile matrix was thus generated for each section. To assign the captured RNAs to individual cells, we transformed the CID-containing gene expression matrix to an image by summing the MID counts at every pixel, and aligned with the corresponding ssDNA image based on patterned track lines in both images. Cell segmentation was performed on the registered ssDNA images by a UNet-like deep convolutional neural network (Supplementary Fig. 8a). To recall the genes from the cytoplasm surrounding the nucleus, a Gaussian mixture model was employed to adjust the cell boundaries. Thus, a single cell-based gene expression matrix was generated for each section by aggregating the spots included in each cell (Supplementary Fig. 8a). To facilitate the analysis of anatomical function of mouse brain, these 3 sections were aligned to the three-dimensional (3D) standard Allen Brain Atlas (Supplementary Fig. 8b) by Wholebrain (https://github.com/tractatus/wholebrain).

## QC and Gaussian smoothing for large field mouse brain hemisphere

The 3 adjacent coronal sections captured 53,310 cells, 61,910 cells, and 61,857 cells, respectively. Pearson coefficient was used to evaluate the consistency between them (Supplementary Fig. 8c). Low-quality cells and genes were discarded according to the following quality control (QC) criteria: (1) cells with total counts lower than 300 and higher than 98% quantile, (2) cells with percentage of mitochondrial genes larger than 10%, (3) genes presented in less than 10 cells. Thus, 41,766 cells, 48,721 cells, 50,329 cells were remained for the 3 adjacent coronal sections (~434 genes per cell), respectively. Next, a Gaussian smoothing strategy was introduced to alleviate the noise and gene dropout

associated with the Stereo-seq technology (Supplementary Fig. 8d). Specifically, the principal component analysis (PCA) was firstly applied cell-wise to reduce the raw gene expression matrix into a low dimension feature space. Then a number of nearest neighboring cells of each cell in the feature space were acquired, which were used to update the gene expression of current cell by Gaussian smoothing.

## Deep neural networks for transfer learning

The definition of transfer learning is that a machine learning model gains problem-solving knowledge from the source domain and stores the knowledge, then the model is applied to solve similar problems in the target domain. The source domain refers to the reference datasets, while the target domain refers to the SRT datasets. In the framework of Spatial-ID, the stage 1 employs the transfer learning strategy to train the deep neural network (DNN) model (Fig. 1a) on reference scRNA-seq datasets that is used to generate the probability distribution $D$ of each cell in SRT data. It should be noted that the DNN model only take the gene expression profiles as input, whereas the available spatial information of SRT data is not used. The DNN provides nonlinear dimensionality reduction for the input gene expression matrix, that consists of 4 stacked fully connected layers, as well as a GELU layer (nonlinear activation function) and a dropout layer followed by each fully connected layer in sequence (see supplementary Table 4). Moreover, to alleviate the class imbalance of cell types, the loss function of Focal Loss is employed in DNN models training.

To perform transfer learning, the gene sets of reference scRNA-seq dataset and SRT dataset should be matched. We first compare the measured genes to find the common gene set. If the reference scRNA-seq datasets or SRT datasets provide marker genes of cell types, we further select the subset of marker genes from the common gene set. This process can simplify the implementation of transfer learning. Then, the raw counts of selected genes from reference scRNA-seq dataset and SRT dataset are normalized to unit length vector for each cell. If there is no marker gene available (e.g., mouse spermatogenesis), we additionally perform a stage of log1p operation to raw counts of genes before the normalization, that intends to inhibit the negative effects of very highly expressed genes.

## Spatial neighbor graph construction for spatial information

To perform spatial embedding, we construct a spatial neighbor graph to represent the spatial relationships between neighboring cells (Fig. 1b). The spatial neighbor graph consists of nodes and edges, where a node represents a cell and an edge represents the relationship of a pair of neighboring cells. To better characterize the relationships, we calculate the Euclidean distance between current cell and neighboring cells using the spatial coordinates. Because the behavior of an individual cell is mediated by the ligand-receptor interactions with its neighboring cells in local tissue microenvironment, so closer distance indicates more closer relationship. For each selected neighbor, we calculate the weight negatively associated with its Euclidean distance by

$$\omega(u,v) = e^{-\frac{d(u,v)^2}{2\theta^2}}, \tag{1}$$

where $d(u,v)$ denoted the Euclidean distance of the coordinates of a pair of neighboring cells/spots in spatial space, $\theta$ denoted the decay coefficient. Specifically, we select the top $N$ nearest neighbors (e.g., 30 in this study) of each cell to create the adjacency matrix, denoted by $A$, in which a cell $u$ with top $N_u$ neighboring cells can be calculated by

$$A(u,v_i) = \omega(u,v_i) \, if \, v_i \in N_u \, else \, 0. \tag{2}$$

## Deep autoencoder for latent representation learning

A deep autoencoder[42] is used to learn encoded gene representation $X$ through reducing the dimension of the gene expression matrix

$I$ (Fig. 1c). If the SRT datasets contains tens of thousands of genes, the gene expression matrix $I$ is generated by extracting hundreds of principal components (e.g., 200) of principal component analysis (PCA). The encoder part consists of 2 stacked fully connected layers, as well as a batch normal layer, a ELU layer (nonlinear activation function) and a dropout layer followed by each fully connected layer in sequence. The decoder part consists of one fully connected layer and same followed layers as encoder. The deep autoencoder employs the mean squared error (MSE) loss function to maximize the similarity between the input gene expression matrix $I$ and the output gene expression matrix $I'$ reconstructed by decoder.

### Variational graph autoencoder for spatial embedding
A GCN[32], i.e., variational graph autoencoder (VGAE)[43], is used to embed spatial neighbor graph (Fig. 1d). Because the spatial neighbor graph contains large number of nodes for high-throughput cell-level SRT data, we employ sparse graph convolution layers in VGAE to accelerate the computation. The variational modification of the encoder-decoder architecture can introduce regularization in the latent space, thus improves the properties of spatial embeddings. The graph encoder takes encoded gene representations $X$ from autoencoder and the adjacency matrix $A$ as input, then generates the spatial embedding $S$ as output. The graph encoder consists of 2 sparse graph convolution layers, as well as a RELU layer (nonlinear activation function) and a dropout layer followed by each graph convolution layer in sequence. The first sparse graph convolution layer is used to generate a lower-dimensional feature matrix. Next, the second sparse graph convolution layers generate a feature matrix $\mu$ and a feature matrix $log\sigma^2$, respectively. Then the spatial embedding $S$ is calculated using parameterization trick $S = \mu + \sigma * \tau$, where $\tau \sim N(0,1)$. The final latent representations $Z$ are combined from the encoded gene representation $X$ and the spatial embedding $S$ by formula $Z = X + S$. Thereafter, the final latent representations $Z$ are used to reconstruct the gene expression matrix $I'$ in the autoencoder and the adjacency matrix $A'$ in the VGAE. Specifically, the graph decoder adapts an inner product to reconstruct adjacency matrix $A'$ from the spatial embedding $Z$. The loss function of the VGAE employs a cross-entropy loss to minimize the input adjacency matrix $A$ and the reconstructed adjacency matrix $A'$, and a KL-divergence to measure the similarity between $q(Z)X, A)$ and $p(Z)$, where $p(Z) \sim N(0,1)$.

### Self-supervised learning using pseudo-labels
We additionally perform a self-supervised learning strategy to train a classifier using the final latent representations $Z$ and pseudo-labels $L$. Essentially, the self-supervised learning employs the pseudo-labels that are generated using the gene expression features of the SRT dataset itself. Formally, the pseudo-labels $L = \{l_1, \ldots, l_i, \ldots, l_n\}$ are derived from the output $Y = \{y_1, \ldots, y_i, \ldots, y_n\}$ of last fully connected layer of the DNN model by a modified softmax layer

$$l_i = \frac{e^{(y_i/T)}}{\sum_j e^{(y_j/T)}}, \quad (3)$$

where $T$ is the temperature parameter[45] that is used to adjust the smoothness of distribution by the temperature setting strategy (Fig. 1b), and $L = D$, if $T = 1$. Obviously, a higher value for $T$ produces softer distribution of pseudo-labels $L$ over all classes. The softer distribution of pseudo-labels $L$ could transfers more information from the reference scRNA-seq datasets.

### Simulation of different gene dropout rates
Many spot-based ST technologies usually have lower mRNA capture efficiency due to the limited number of probes. Therefore, the gene dropout rates of spot-based SRT datasets are usually higher than the state-of-the-art scRNA-seq technologies and FISH-based SRT

technologies. To verify the adaptability of Spatial-ID in different gene dropout rates, we conduct simulation experiments on FISH-based mouse primary motor cortex SRT dataset (Fig. 2j). The simulated SRT datasets are generated by a random-gene-discard strategy, that randomly resets part of values to 0 in the gene expression matrix. Then, we compare cell type annotation of Spatial-ID and control methods on these simulated SRT datasets.

### New cell type discovery
The discovery of new cell populations has biologically important implication for omics analysis. Technically, the new cell type discovery problem can be characterized as an anomaly detection task. For example, the reference datasets do not contain the new cell types to be found in SRT datasets or have serious class imbalance problem derived from scarce cells of the new cell types, thus these new cell types are identified into incorrect cell types (e.g., L4/5 IT and L6 IT car3 in mouse primary motor cortex dataset). For another situation, the SRT datasets have indistinguishable feature distributions (e.g., the indistinguishable cell types in human NSCLC dataset).

Essentially, the new cell type discovery task is performed on the cell type probability distribution outputted by cell typing methods. It is a postprocess of cell typing methods that attempts to identify new categories from low-confidence cells. As shown in Fig. 2k, the process of new cell type discovery contains thresholding, clustering, mapping, and filtering sub-steps. For each cell, the maximum score $c_s \in [0,1]$ of predicted probability vector $\mathbf{C} = \{c_1, \ldots, c_i, \ldots, c_{n-1}\}, \sum_1^{n-1} c_i = 1$ is first examined by a threshold at the thresholding step. If the maximum score $c_s$ of the cell is less than the threshold (e.g., 0.9 in Fig. 2k), the cell is tagged as an unassigned cell. The lower threshold retrieves fewer unassigned cells, that makes the subsequent new class discovery process impossible to find new cell type patterns. On the other hand, the higher threshold retrieves more unassigned cells, that leads more clusters to filter in the subsequent analysis. The threshold is empirically set in the range of 0.5 to 0.9. We set a threshold of 0.9 for demonstration in Fig. 2k and a threshold of 0.7 for demonstration in Fig. 5e–h. The clustering step is performed to group the unassigned cells into different clusters, that could ensure that the new cell types populate in a relatively isolated area in the feature space. Empirically, the number of clusters approximately equals to the half of total cell types. Next, the mapping step aligns the clusters to predicted cell types in the feature space for these unassigned cells (Supplementary Fig. 2g and Fig. 5e–g). Each cluster may contain multiple cell types predicted by Spatial-ID. For example, the cluster 2 in Supplementary Fig. 2g contains the L2/3 IT and L5 IT cell types predicted by Spatial-ID, and the gene expression profile of this cluster presents gradual transition between the L2/3 IT and L5 IT cell types. The filtering step analyzes the distance from the center of each cluster to the center of the predicted types one-by-one. If the distance is less than the radius of one predicted type in feature space, the cluster is still identified as this predicted type rather than a new cell type. If the distance is larger than the radius of all predicted types in the feature space, the cluster is identified as a new cell type.

### Identifying spatial gene patterns for large field mouse brain hemisphere
Hotspot[67] is used to identify spatial gene patterns (Supplementary Fig. 9b). First, data binning is performed to reduce computational difficulty by dividing the x-y coordinates into grids covering an area of 50*50 DNB (bin50) and the transcripts of the same gene aggregated within each bin. Next, highly variable genes are found by toolkit scanpy[75]. Hotspot uses the number of 300 neighbors to create the spatial KNN graph, and then uses spatially-varying genes (FDR <= 0.05) to identify spatial gene patterns. In addition, clusterProfiler[76] is used to perform GO enrichment analysis of the identified spatial gene patterns (Fig. 6j).

## Computational resources and runtime

All analyses presented in the paper are run in a workstation with 40 Gb RAM memory, 10 cores of 2.5 GHz Intel Xeon Platinum 8255C CPU, and a Nvidia Tesla T4 GPU with 8 Gb memory. And the following python (v3.8) packages support for Spatial-ID are required: numpy==1.21.3, pandas==1.2.4, scipy==1.5.4, matplotlib==3.3.4, seaborn==0.11.1, scikit-lean==0.24.2, torch==1.8.1, torch_geometric==1.7.2, scanpy==1.8.1. The runtime of different cell typing methods in this workstation are shown in Supplementary Table 3.

## Statistics and reproducibility

A summary of all involved datasets is given in Supplementary Table 1. The four publicly available SRT datasets (i.e., mouse brain primary motor cortex, mouse brain hypothalamic preoptic region, mouse spermatogenesis, and human non-small-cell lung cancer) contained 12, 3, 6, 20 samples, respectively. The self-collected mouse brain hemisphere dataset contained one sample with 3 slices. No sample was excluded in this study. All boxplots in this study, draw by matplotlib and seabron python package, employed the same settings to show the median (center lines), the inter-quantile range (IQR, the 25th and 75th percentiles for the lower and the upper hinges, respectively) and the whiskers (extend up to 1.5 IQRs from the lower and upper hinges, respectively). The points that fell outside the whiskers were "outliers" and were displayed independently. Wilcoxon rank-sum test was calculated by scipy python packge to statistically analyze differences between methods. The regression plots, draw by matplotlib and seabron python package, were presented as mean values with 95% confidence intervals.

The visualizations of ground truths in Figs. 2b, d, h, k, 3b, d, 4a, d, e, 5b, c and Supplementary Figs. 1a–c, 2e, 5d, 6a, b were generated from the annotations provided by publicly available datasets, which can be reproduced infinitely. The results of Spatial-ID and control methods in Figs. 2c, e, h, j, k, 3c, d, 4b, d, e, 5d–h, 6b, and Supplementary Figs. 1a–c, 2f, g, 5d, 6a, b, 8a were similar with at least 3 independent experiments replicates.

## Reporting summary

Further information on research design is available in the Nature Portfolio Reporting Summary linked to this article.

## Data availability

A summary of all involved datasets is given in Supplementary Table 1. The public datasets are freely available as follow. Mouse brain - primary motor cortex (MOP): "https://doi.org/10.35077/g.21 [https://doi.brainimagelibrary.org/doi/10.35077/g.21]". Mouse brain - hypothalamic preoptic region: "https://doi.org/10.5061/dryad.8t8s248 [https://datadryad.org/stash/dataset/doi:10.5061/dryad.8t8s248]". "Mouse spermatogenesis [https://www.dropbox.com/s/ygzpj0d0oh67br0/Testis_Slideseq_Data.zip?dl=0]". Human NSCLC: "SMI-FFPE Dataset–Lung9-Rep1 Data [https://nanostring.com/resources/smi-ffpe-dataset-lung9-rep1-data/]". The snRNA-seq 10x v3 B of BICCN MOP dataset (RRID: SCR_015820) can be accessed via the NeMO archive (RRID: SCR_002001) at "accession [https://assets.nemoarchive.org/dat-ch1nqb7]". scRNA-seq of preoptic region of mouse hypothalamic: "GSE113576". scRNA-seq of mouse testis: "GSE112393". "scRNA-seq NSCLC [https://gbiomed.kuleuven.be/english/research/50000622/laboratories/54213024/scRNAseq-NSCLC]". Mouse brain atlas of cell types from the Linnarsson Lab: "SRP135960 [http://mousebrain.org/adolescent]". The raw Stereo-seq sequencing data used in this study have been deposited in China National Gene Bank (CNGB) Sequence Archive (accession code: "CNP0002966"), that are available from the corresponding author upon reasonable request. The raw and processed Stereo-seq sequencing data have been deposited in Zenodo (https://doi.org/10.5281/zenodo.7340795)[77] that are publicly accessible. All other relevant data

supporting the key findings of this study are available within the article and its Supplementary Information files or from the corresponding author upon reasonable request. Source data are provided with this paper.

## Code availability

An open-source Python implementation of Spatial-ID and reproduction code are available at https://github.com/TencentAILabHealthcare/spatialID and Zenodo[78] (https://doi.org/10.5281/zenodo.7315186).

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

## Acknowledgements

Thanks to the following organizations for Stereo-seq data collection: Guangdong Provincial Key Laboratory of Genome Read and Write (No. 2017B030301011), China National GeneBank.

## Author contributions

Conceptualization: Rongbo Shen, Lin Liu, Zihan Wu, and Ying Zhang. Project administration and supervision: Yuxiang Li, Xun Xu, and Jianhua Yao. Algorithm development and implementation: Zihan Wu and Rongbo Shen. Public datasets collection, processing and application: Rongbo Shen, Zihan Wu, Wanwan Feng, and Zhiyuan Yuan. Stereo-seq dataset collection, processing and application: Rongbo Shen, Zihan Wu, Lin Liu, Ying Zhang, Junfu Guo, Chao Zhang, Bichao Chen, Chao Liu, Jing Guo, Guozhen Fan, Yong Zhang. Methods comparisons: Rongbo Shen and Zihan Wu. Biological interpretation: Lin Liu, Rongbo Shen, and Ying Zhang. Project coordination: Fang Yang and Bichao Chen. Manuscript writing and figure generation: Rongbo Shen and Lin Liu. Manuscript reviewing: Zhiyuan Yuan, Fang Yang, and Jianhua Yao. Major revision: Rongbo Shen, Zihan Wu, and Lin Liu. Final revision: Rongbo Shen. All authors approved the manuscript.

## Competing interests

The authors declare no competing interests.
