## [Peer Review File · Nature Communications]

Spatial-ID: a cell typing method for spatially resolved transcriptomics via transfer learning and spatial embeddingREVIEWER COMMENTS

Reviewer #1 (Remarks to the Author):

In this manuscript, Shen et al. present a novel supervised cell type annotation method for spatial transcriptomics that utilizes spatial information. The authors highlight the superior performance of the method on 3 datasets, showing Spatial-ID's better accuracy, runtime efficiency, and robustness to dropout. While the results are generally convincing, resolving points #1 and #2 below in particular will add substantially to the strength of the manuscript.

Major revisions:

1. The authors include several transfer learning and other supervised cell typing methods in their benchmark. However, most of these were developed for scRNAseq, while Tangram (Biancalani et al., 2021) is a label transfer method designed for SRT technologies and would better represent the state of the art method. This method should be included in the benchmarking analyses.
2. The results in Figure 2K seem a bit cherry-picked, given that there are many other cells "unassigned" besides those that lie in the ground truth L4/5IT and L6 IT Car3 clusters. There is not enough discussion on how the thresholding is done and how a user is expected to conduct cell typing on the other (7) unsupervised clusters shown within the "Clustering" panel. Do those other clusters also correspond well with ground truth labels? Semi-supervised clustering is a challenging problem and the current approach that the authors describe seems more like a teaser than a full solution.
 - a. Not required, but I think an immune/tumor analysis would really highlight the versatility of the method across different datasets and strengthen the paper. It is typically hard to find good references for tumor, so tumor samples would test the performance of this "new cell type discovery" is.
3. In the discussion, the authors briefly touch on the effect of hyperparameters, though there is relatively little elaboration on how these hyperparameters were chosen and the effect of these on end results. Will users of the method be expected to tune these hyperparameters. In particular, the authors mention in the Intro that they were inspired by other methods that took into account spatial information, so some further results showing the effect of tuning the spatial smoothing would be nice.
4. While code has been provided, there is very limited documentation and only analysis scripts rather than a module/package. Without additional packaging and documentation, I think it will be difficult for users to run this method on their own data.

Minor suggestions:

1. A little more elaboration of why we need Stage 3 after Stage 2 would be useful in the Results section. Why aren't the results from Stage 2 sufficient? Explain the intuition for why we need to use this self-supervised approach.
2. In the writeup of results for Fig. 4, the text does not introduce the panels in alphabetical order (e.g. 4e discussed before 4c). 4b is also not mentioned.
3. The description of the UMAP as separating cell types into "separable communities" seems a bit misleading given that most clusters aren't actually separating.
4. Line 426, should introduce Hotspot (briefly) before talking about results from it.
5. Line 668, this description is not clear to me. Maybe provide an example of how this is done.

Reviewer #2 (Remarks to the Author):

The authors present Spatial-ID a novel method for annotating cells or spots in cellular or subcellular resolution spatial transcriptomic data (e.g. Stereo-seq) that combines: a deep neural network trained on a reference single-cell or single-nucleus RNAseq dataset which generates labels that are further refined using an autoencoder-driven embedding that combines the transcriptomic data from the spatial method with the spatial location of cells via a kNN graph. This is compared to several methods of reference-based autoannotation of single-cell or single-nucleus RNAseq data.

While the method is novel and shows improvement over the non-spatial methods, the authors do not compare their tool to existing tools designed to annotate and deconvolve spatial transcriptomics data. The manuscript also has insufficient description of their method to enable reproduction of their results

particularly with respect to hyper-parameters.

Major :

Spatial ID is much more similar in concept to spatial deconvolution methods - using a well annotated reference single-cell dataset to label spatial transcriptomics data either with a single label per spot or with a set of cell-type proportions per spot - but the introduction primarily discusses single-cell cell-type annotation tools. Likewise, it should be compared to spatial deconvolution tools in addition to or instead of single-cell automatic annotation tools.

Spatial-ID claims the improvement in its accuracy is derived from inclusion of spatial location information, yet this is incorporated only in a small portion of the overall model. How much of an improvement in accuracy is truly derived from the addition of spatial location? Could the authors repeat their benchmarks with their method excluding the adjacency matrix to quantify the benefits of incorporating spatial location information?

Results:

It's unclear where the ground truth for the cell/spot labelling came from that the methods were benchmarked against. Were the author reported computationally-derived annotations used? Was it manually annotated?

It should be made clear throughout the manuscript that a distinct pipeline is necessary for novel cell-type discovery that is not inherent to the Spatial-ID pipeline (Figure 1 k). In the introduction and abstract it is made to sound like Spatial-ID simultaneously finds novel cell-types and identified reference cell-types but this is not the case. In addition, the authors should comment on the fact that the novel cell type pipeline identified 8 novel clusters when then expected only 2 - could they identify the remaining 6 novel clusters as truly novel cell-types as well?

Methods:

There is a lack of specifics of the "novel cell type pipeline" - how the threshold for unassigned is determined, what clustering method was applied and what the parameters were set at and to what representation of the data the clustering was applied. The methods must be updated to include all of this information so that others can reproduce the author's work.

Performance of neural networks and autoencoders tend to be sensitive to their hyperparameters. A full list of all hyperparameters and their values for each dataset analyzed in the manuscript must be provided in the Methods section.

Minor:

Seurat v3 is misspelled & repeated at line 84-85

"newly sequencing dataset" - sequenced at line 95, 99, 105

"PGM-based" - PGM needs to be defined.

cell2location is a leading spatial deconvolution method that should be mentioned.

"self-collected Stereo-seq" (line 183) - did you mean newly generated Stereo-seq dataset?

Introduction does not make it clear why the ability to cluster spatial transcriptomic data would imply spatial data would be useful for cell-type annotation. Description of the approach in the introduction (line 124-137) is unclear what the output of the method is and how the various algorithms arrive at that output - what dataset is it labelling? How are those labels arrived at?

Reviewer #3 (Remarks to the Author):

This paper deals with the development of a method ("Spatial-ID") that can classify single cells analyzed in the context of spatially resolved transcriptomics (SRT). SRT is an important technique that

does not only provide a snapshot of the gene expression status of a single cell (like ordinary scRNA-seq), but also provides information on the spatial location of cells relative to other cells analyzed.

The authors present a transfer learning based framework by which to determine the type of cells from SRT data sets. In more detail, the authors first train a deep neural network to generate labels (termed "pseudo-labels") using known reference data sets. Then, they classify the single cells from the SRT data set using the deep neural network, that is, they assign "pseudo-labels" to each single cell from the SRT data set. Subsequently, the pseudo-labels are used to train a graph neural network / autoencoder / variational graph autoencoder based neural network part that is concerned with learning appropriate embeddings and representations of the SRT data. When done training, the corresponding neural network part is used for final classification.

The theoretical part appears sound to me; however, the authors do not provide various details (definitions, parameters etc) that enable one to fully reproduce the architectures of the networks. Results demonstrate the superiority of Spatial-ID very convincingly. However, Results refer to mouse data sets alone, which has left me with some doubts. The analysis of additional data sets may be helpful to fully convince the reader of the universal benefits of the approach.

In summary, this appears to be a very promising paper to me, which however leaves a few important questions unanswered so far.

MAJOR:

* It would be appropriate to provide a concise, short summary of the open challenges remaining in spatially resolved transcriptomics, together with a description of advances of Spatial-ID that addresses the open challenges the authors have identified. The authors only list related work with the corresponding brief descriptions of the tools. Then they describe what Spatial-ID does. However, I am missing to put the deficiencies of earlier work into context with the advances Spatial-ID brings up. To be more specific: are there no methods that combine prior knowledge based on existing reference scRNA-seq datasets on the one hand, and spatial information retrieved from SRT datasets on the other hand?

This remains somewhat unclear.

* "the DNN model of transfer learning generates the probability distribution of each cell" -> which probability distribution do you mean? Please clearly define "the probability distribution of a cell".

* What is a "pseudo-label"? Please define/explain.

* What is a "temperature setting strategy of knowledge transfer"? Please explain, and provide suitable citations.

* How is the "Euclidean distance" between cells defined? In other words, what is the definition of a "cell" here?

* All of the first paragraph after "The pipeline of Spatial-ID" is quite unclear for a reader, even if certainly knowledgeable in neural networks (which I am). Likely, a reader not knowledgeable in neural networks is entirely lost. It would be favorable to explain the workflow in more detail, for example, by making more accurate use of the figures that illustrate it. Also, one has to provide formal definitions for all concepts that are used in a formal context. In particular, what is a "cell", as an input for the neural networks in use here?

* In Figure 1, what is 'CLS' supposed to mean? I was not able to find a definition throughout the entire text of the manuscript.

* Is it possible to run Spatial-ID on a dataset that does *not* refer to mouse cells? The experiments provide the reader with the idea that the method predominantly works for mouse. Experiments on an additional species would be very convincing. If this is not possible, for example because there are no data available, please point this out and discuss the possible range of applications in the future.

* Later, in Methods, the crucial details and parameters that describe the DNN, and also the GCN, are missing entirely. This means that the Methods remain irreproducible. Please provide a sufficiently detailed description of the architectures of all networks involved in the Methods part.

* Also, the descriptions of the autoencoders lack necessary details, such as sizes of layers, and so on. Maybe an appropriate figure can do the job.

* Referring to the definition of "pseudo-label" as well, I believe that is a good idea to provide a brief introduction to transfer learning, such that a reader not familiar with the basic concept can follow the paper. For example, just briefly point out what a "teacher" does, and what a "student" does. Then, everything should be clear.

MINOR:

* Seruat -> Seurat

* There are some minor edits to be done with respect to language, such as missing articles, and similar things. This is not so important, because I find the manuscript already perfectly readable.

Dear Reviewers:

We would like to thank you for your careful reading, helpful comments, and constructive suggestions. These comments are all valuable for revising and improving the presentation of our manuscript, as well as the important guidance for our researches. We have studied all comments carefully and have made correction which we hope to sufficiently answer reviewers' concerns. The main corrections in the study and the responds to the reviewer's comments are as follow.

REVIEWER COMMENTS

Reviewer #1 (Remarks to the Author):

In this manuscript, Shen et al. present a novel supervised cell type annotation method for spatial transcriptomics that utilizes spatial information. The authors highlight the superior performance of the method on 3 datasets, showing Spatial-ID's better accuracy, runtime efficiency, and robustness to dropout. While the results are generally convincing, resolving points #1 and #2 below in particular will add substantially to the strength of the manuscript. **Reply: Thanks for your valuable comments. We really appreciate your efforts in reviewing our manuscript. We have revised the manuscript accordingly. Our point-by-point responses are detailed below. The revised portion are marked in red in each response.**

Major revisions:

1. The authors include several transfer learning and other supervised cell typing methods in their benchmark. However, most of these were developed for scRNA-seq, while Tangram (Biancalani et al., 2021) is a label transfer method designed for SRT technologies and would better represent the state of the art method. This method should be included in the benchmarking analyses.

Reply: Thanks for your suggestion. We add Tangram in the benchmarking analysis and report the performance of Tangram in the revised manuscript. On the four public datasets, our proposed Spatial-ID achieves superior performance than Tangram, including accuracy, weighted F1 score, runtime efficiency. Notably, apart from the 3 public datasets used in the original manuscript, we add a Human NSCLC (non-small-cell lung cancer) dataset in the benchmarking analysis. The accuracy of each sample from the 4 datasets is shown in the following tables, which is also visualized for comparison in Fig. 2f, Fig. 3e, Fig. 4c, Fig. 5a. Moreover, the comparison of weighted F1 score can be found in Extend Fig. 2a, Extend Fig. 7a, Extend Fig. 7b, Extend Fig. 7c. The comparison of runtime efficiency can be found in Table 3. The robustness test of Tangram can be found in Fig. 2j and Extend Fig. 5a.

Sample	Spatial-ID	Tangram
mouse1_sample1	91.20%	76.40%
mouse1_sample2	92.08%	81.94%
mouse1_sample3	92.92%	84.16%
mouse1_sample4	91.36%	84.28%
mouse1_sample5	92.26%	86.24%
mouse1_sample6	92.99%	88.88%
mouse2_sample1	92.85%	84.19%
mouse2_sample2	93.66%	86.70%
mouse2_sample3	93.58%	85.71%
mouse2_sample4	93.43%	86.74%
mouse2_sample5	93.25%	84.58%
mouse2_sample6	93.42%	86.83%

Table 1-1 Mouse brain - primary motor cortex (MOP) dataset. Mean accuracy for Spatial-ID and Tangram: 92.75%, 84.72%.

Sample	Spatial-ID	Tangram
sample1	86.94%	58.62%
sample2	87.51%	57.64%
sample3	88.76%	61.20%

Table 1-2 Mouse brain – hypothalamic preoptic region (3D) dataset. Mean accuracy for Spatial-ID and Tangram: 87.74%, 59.15%.

Sample	Spatial-ID	Tangram
diabetes_1	59.66%	57.86%
diabetes_2	59.26%	54.60%
diabetes_3	62.84%	60.11%
wild_type_1	55.66%	55.89%
wild_type_2	55.61%	56.44%
wild_type_3	69.82%	64.30%

Table 1-3 Mouse spermatogenesis dataset. Mean accuracy for Spatial-ID and Tangram: 60.45%, 58.20%.

Sample	Spatial-ID	Tangram
1	37.32%	19.13%
2	48.91%	22.97%
3	35.29%	18.80%
4	36.59%	19.00%
5	94.22%	10.37%
6	72.04%	13.70%
7	52.86%	28.73%
8	68.80%	16.71%
9	87.83%	12.70%
10	74.50%	19.33%
11	62.08%	17.20%
12	81.23%	10.29%
13	81.29%	17.64%
14	86.91%	14.69%
15	74.90%	13.55%
16	77.58%	15.25%
17	82.38%	15.59%
18	86.15%	12.65%
19	66.95%	13.38%
20	87.08%	10.80%

Table 1-4 Human NSCLC (non-small-cell lung cancer) dataset. Mean accuracy for Spatial-ID and Tangram: 69.76%, 16.12%.

Fig. 2f and Fig. 3e show the accuracy performance for mouse primary motor cortex (MOP) dataset and mouse hypothalamic preoptic region (3D) dataset.

Fig. 4c and Fig. 5a show the accuracy performance for mouse spermatogenesis dataset and human NSCLC dataset.

Extend Fig. 2a and Extend Fig. 7a show the weighted F1 score for mouse primary motor cortex (MOP) dataset and mouse hypothalamic preoptic region (3D) dataset.

Extend Fig. 7b and Extend Fig. 7c show the weighted F1 score for mouse spermatogenesis dataset and human NSCLC dataset.

Fig. 2j and Extend Fig. 5a show for robustness test of Spatial-ID and the control methods on the mouse primary motor cortex (MOP) dataset. Left: accuracy; Right: weighted F1 score.

We also add a description of Tangram in the introduction section as follow.

Line 129 to 132 in revised manuscript:

Tangram employed a deep learning framework based on nonconvex optimization to align scRNA-seq data to various forms of spatial data collected from the same tissue, then mapped the cell types defined by scRNA-seq on the spatial context.

2. The results in Figure 2K seem a bit cherrypicked, given that there are many other cells “unassigned” besides those that lie in the ground truth L4/5IT and L6 IT Car3 clusters. There is not enough discussion on how the thresholding is done and how a user is expected to conduct cell typing on the other (7) unsupervised clusters shown within the

“Clustering” panel. Do those other clusters also correspond well with ground truth labels? Semi-supervised clustering is a challenging problem and the current approach that the authors describe seems more like a teaser than a full solution.

a. Not required, but I think an immune/tumor analysis would really highlight the versatility of the method across different datasets and strengthen the paper. It is typically hard to find good references for tumor, so tumor samples would test the performance of this “new cell type discovery” is.

Reply: Thanks for your comment. To make the process of new cell type discovery clearer, we describe more details of the analysis in the manuscript. Besides, we add a Human NSCLC (non-small-cell lung cancer) dataset and perform the process of new cell type discovery.

Essentially, the new cell type discovery is performed on the cell type probability distribution outputted by cell typing methods. It is a postprocess of cell typing methods that attempts to identify new categories from low-confidence cells.

As shown in Fig 2k, the process of new cell type discovery contains Thresholding, Clustering and Mapping & Filtering sub-steps.

- (1) Thresholding:** the maximum score $c_s \in [0, 1]$ of predicted cell type probability distribution of each cell is first examined by a threshold. If the maximum score c_s of a cell is less than the threshold, the cell is tagged as an unassigned cell. The lower threshold retrieves less unassigned cells, that makes the subsequent new class discovery process impossible to find new cell type patterns. On the other hand, the higher threshold retrieves more unassigned cells, that leads more clusters to filter in the subsequent analysis. The output layer of classifier in Spatial-ID is a softmax function, therefore the cells that are easier to classify usually have a maximum score close to 1. The threshold is empirically set in the range of 0.5 to 0.9. We set the threshold of 0.9 for demonstration in Fig. 2k.
- (2) Clustering:** The clustering step is performed to group the unassigned cells into different clusters. Empirically, the number of clusters approximately equals to the half of total cell types. For example, the reference dataset contains 20 cell types. In Fig. 2k, the unassigned cells are clustered into 9 clusters.
- (3) Mapping:** According to the clusters, the mapping step aligns the clusters of unassigned cells to the predicted cell types in the same feature space, as shown in Extend Fig. 2g. Compared with the prediction of Spatial-ID (Extend Fig. 2f), each cluster contains several types predicted by Spatial-ID. For example, the cluster 2 contains the L2/3 IT and L5 IT cell types predicted by Spatial-ID, the cluster 5 contains the L6 IT cell type predicted by Spatial-ID.
- (4) Filtering:** The distance from the center of each cluster to the center of the predicted types is analyzed. If the distance is less than the radius of one predicted type in the feature space, the cluster is still identified as this predicted type rather than a new cell type. If the distance is larger than the radius of all predicted types in feature space, the cluster is identified as a new cell type. For example, the distance of cluster 2 to L2/3 IT cell types predicted by Spatial-ID in

feature space is larger than the radius of predicted L2/3 IT cell types, as well as L5 IT cell types predicted by Spatial-ID. Therefore, the cluster 2 is identified as a new cell type. This new cell type is labelled as L4/5 IT in ground truth (Extend Fig. 2e). Compared with the ground truth, we can calculate the recall 0.258, precise 0.945 and F1 score 0.405 for this new cell type. Similarly, the cluster 5 is identified as a new cell type, that is labelled as L6 IT car3 in ground truth. Apply the above filtering rule to the other clusters (0,1,3,4,6,7,8) in turn, we can find that these clusters are closer to their main cell type predicted by Spatial-ID, therefore these clusters are not identified as new cell types. For example, the distance of cluster 0 to L6 IT is too closer, so this cluster is still identified as L6 IT. The distance of cluster 4 to L2/3 IT is too closer, so this cluster is still identified as L2/3 IT.

Fig. 2k

Extend Fig. 2f

Extend Fig. 2g

For the Human NSCLC (non-small-cell lung cancer) dataset, we also perform new cell type discovery after Spatial-ID, as shown in Fig. 5e-h. The process of new cell type discovery can retrieve the cell types of Neutrophil and Endothelial that are missed in the prediction of Spatial-ID (Fig. 5d). Because the Human NSCLC dataset has indistinguishable feature distributions for these scarce cell types.

Fig. 5e-h

Line 320 – 352 in revised manuscript for MOP dataset:

L4/5 IT and L6 IT Car3 neurons are not annotated in the snRNA-seq 10x v3 B dataset, thus Spatial-ID predicts the L6 IT Car3 neurons as L6 IT neurons, and L4/5 IT neurons as L2/3 IT and L5 IT neurons (Fig. 2c and Fig. 2e). The ground truth of MOP dataset shows that L4/5 IT neurons populate in the continuous region between L2/3 IT and L5 IT neurons (Fig. 2d), and L6 IT Car3 neurons populate near the L6 IT neurons at L6 layer (Fig. 2d). These results suggest that the gene expression profiles of L4/5 IT neurons present gradual transition between the L2/3 IT and L5 IT neurons, and the characteristics of L6 IT Car3 neurons are similar to the L6 IT neurons (Fig. 2c). We conduct a postprocess for Spatial-ID to further distinguish these new cell types that are presented in the MOP dataset but unseen in the snRNA-seq 10x v3 B dataset. Based on the predictions of Spatial-ID, a pipeline of new cell type discovery (Fig. 2k) is

performed to determine whether there are new cell types in the MOP dataset, including thresholding, clustering, mapping and filtering (See Methods). The unassigned cells are determined by a threshold (i.e. 0.9) at the thresholding step. The clustering step is performed to group the unassigned cells into different clusters (i.e. 9 clusters). According to the clusters, the mapping step aligns the clusters of unassigned cells to predicted cell type in the same feature space, as shown in Extend Fig. 2g. Compared with the predictions of Spatial-ID (Extend Fig. 2f), each cluster contains several cell types predicted by Spatial-ID. For example, the cluster 2 contains the L2/3 IT and L5 IT cell types predicted by Spatial-ID, the cluster 5 contains the L6 IT cell type predicted by Spatial-ID. The filtering step analyzes the distance from the center of each cluster to the center of the predicted cell types. If the distance is less than the radius of one predicted cell type in feature space, the cluster is still identified as this predicted cell type rather than a new cell type. If the distance is larger than the radius of all predicted cell types in the feature space, the cluster is identified as a new cell type. For example, the distance of cluster 2 to L2/3 IT cell types predicted by Spatial-ID in feature space is larger than the radius of predicted L2/3 IT cell types, as well as L5 IT cell types predicted by Spatial-ID (Extend Fig. 2f and Extend Fig. 2g). Therefore, the cluster 2 is identified as a new cell type that is labelled as L4/5 IT in ground truth (Extend Fig. 2e). Similarly, the cluster 5 is identified as a new cell type, that is labelled as L6 IT car3 in ground truth (Extend Fig. 2e). Applying the above filtering rule to the other clusters in turn, we can find that these clusters are closer to their main cell type predicted by Spatial-ID, therefore these clusters are not identified as new cell types. Finally, the discovered L4/5 IT and L6 IT Car3 neurons achieves F1 score of 0.405 and 0.904, respectively (Fig. 2k).

Line 452 – 463 in revised manuscript for Human NSCLC dataset:

The thresholding step determines a set of unassigned cells by a threshold 0.7 (Fig. 5f). Then, these unassigned cells are grouped into 5 clusters at the clustering step (Fig. 5g). Next, the mapping step aligns the clusters to predicted cell types in the feature space for these unassigned cells (Fig. 5g and Fig. 5e). Finally, the filtering step analyzes the distance from the center of each cluster to the center of the predicted cell types. The cluster 0 and cluster 4 are identified as new cell types (Fig. 5h). Obviously, there are labelled as neutrophil cells and endothelial cells in ground truth.

Line 805-840 in Methods section of revised manuscript:

The discovery of new cell populations has biologically important implication for omics analysis. Technically, the new cell type discovery problem can be characterized as an anomaly detection task. For example, the reference datasets do not contain the new cell types to be found in SRT datasets or have serious class imbalance problem derived from scarce cells of the new cell types, thus these new cell types are identified into incorrect cell types (e.g., L4/5 IT and L6 IT car3 in mouse primary motor cortex dataset). For another situation, the SRT datasets have indistinguishable feature distributions (e.g., the indistinguishable cell types in human NSCLC dataset).

Essentially, the new cell type discovery task is performed on the cell type probability distribution outputted by cell typing methods. It is a postprocess of cell typing methods that attempts to

identify new categories from low-confidence cells. As shown in Fig 2k, the process of new cell type discovery contains thresholding, clustering, mapping and filtering sub-steps. For each cell, the maximum score $c_s \in [0,1]$ of predicted probability vector $C = \{c_1, \dots, c_i, \dots, c_{n-1}\}$, $\sum_1^{n-1} c_i = 1$ is first examined by a threshold at the thresholding step. If the maximum score c_s of the cell is less than the threshold (e.g., 0.9 in Fig. 2k), the cell is tagged as an unassigned cell. The lower threshold retrieves fewer unassigned cells, that makes the subsequent new class discovery process impossible to find new cell type patterns. On the other hand, the higher threshold retrieves more unassigned cells, that leads more clusters to filter in the subsequent analysis. The threshold is empirically set in the range of 0.5 to 0.9. We set a threshold of 0.9 for demonstration in Fig. 2k and a threshold of 0.7 for demonstration in Fig. 5e-h. The clustering step is performed to group the unassigned cells into different clusters, that could ensure that the new cell types populate in a relatively isolated area in the feature space. Empirically, the number of clusters approximately equals to the half of total cell types. Next, the mapping step aligns the clusters to predicted cell types in the feature space for these unassigned cells (Extend Fig. 2g and Fig. 5e-g). Each cluster may contain multiple cell types predicted by Spatial-ID. For example, the cluster 2 in Extend Fig. 2g contains the L2/3 IT and L5 IT cell types predicted by Spatial-ID, and the gene expression profile of this cluster presents gradual transition between the L2/3 IT and L5 IT cell types. The filtering step analyzes the distance from the center of each cluster to the center of the predicted types one-by-one. If the distance is less than the radius of one predicted type in feature space, the cluster is still identified as this predicted type rather than a new cell type. If the distance is larger than the radius of all predicted types in the feature space, the cluster is identified as a new cell type.

3. In the discussion, the authors briefly touch on the effect of hyperparameters, though there is relatively little elaboration on how these hyperparameters were chosen and the effect of these on end results. Will users of the method be expected to tune these hyperparameters. In particular, the authors mention in the Intro that they were inspired by other methods that took into account spatial information, so some further results showing the effect of tuning the spatial smoothing would be nice.

Reply: Thanks for your suggestion. We summarize all the hyperparameters in Table 4. For a new application, the hyperparameters of neural network architecture do not need to be tuned. Users only need to modify the input dimension of DNN and the input dimension of the autoencoder (If PCA was applied, the input dimension should be set as the PCA dimension, e.g., 200). In the other hyperparameters, decay coefficient for edge weight is usually set as the pixels of 2 cell diameter in spatial. Users only need to modify the number of neighbor cells.

The number of neighbor cells indicates the use of spatial information. In the benchmarking analysis of the 4 publicly datasets, this hyperparameter is set as 30 consistently. We introduce a set of experiments to compare the effect of this hyperparameter on the hypothalamic preoptic region dataset (Extend Fig. 7e). We can observe that the number of neighbor cells slightly affects the performance of

accuracy and the larger number of neighbor cells may lead to reduced differences between local cells.

Extend Fig. 7e

	Parameters	Value	Explanation
Data preprocess	cell_min_counts	300 (mouse brain hemisphere dataset)	Cells with total counts fewer than cell_min_counts will be removed.
	cell_max_counts_pct	98% (mouse brain hemisphere dataset)	Cells with total counts higher than cell_max_counts_pct quantile will be removed.
	filter_mt	10% (mouse brain hemisphere dataset)	Cells with percentage of mitochondrial genes larger than 10% will be removed.
	gene_min_cells	10 (mouse brain hemisphere dataset)	Genes presented in fewer than 10 cells will be removed.
DNN	Fully connected layer	4	Each fully connect layer is followed by a GRLU layer and dropout layer. Neurons in a fully connected layer have connections to all neurons in the previous layer. 4 fully connect layers contain 128, 64, 64, cell_types neurons for the mouse primary motor cortex dataset, mouse hypothalamic preoptic region dataset and human NSCLC dataset. 4 fully connect layers contain 512, 256, 256, cell_types neurons for the mouse spermatogenesis dataset and mouse brain hemisphere dataset.
	GELU layer	4	A type of activation function layer.
	Dropout layer	4	At training stage, individual neurons are either "dropped out" of the network (ignored) with probability p . Dropout layer is used to reduce overfitting.
	Loss function	Focal Loss	A modified cross entropy loss function is used to alleviate the class imbalance problem.
Autoencoder	pca_dim	None	PCA dimension. For the mouse spermatogenesis dataset and the mouse brain hemisphere dataset, this parameter is set as 200. Other datasets use all the shared genes.
	feat_dim	64	The encoded feature dimension.
	Fully connected layer	2	2 fully connect layers contain 100, 64 neurons, respectively. Each fully connect layer is followed by a batch normal layer, a ELU layer and a dropout layer.
	Batch normal layer	2	Batch normal layer is used to make training of neural networks faster and more stable through normalization of the layers' inputs by re-centering and re-scaling.
	ELU layer	2	A type of activation function layer.
	Dropout layer	2	As above of dropout layer.
	Decoder layer	1	The decoder layer is also a fully connected layer. Neurons in this layers equal to the input dimension (e.g., pca_dim or number of shared genes).
Variational graph autoencoder	sparse GCN layer	2	A GCN layer receives feature vector of each node and neighbor graph (adjacency matrix). GCN layers constrains 32, 8 neurons, respectively. Each GCN layer is followed by a ReLU layer and a dropout layer.
	RELU layer	2	A type of activation function layer.
	Dropout layer	2	As above of dropout layer.
	Inner-product layer	1	Inner-product layer is used to reconstruct the adjacency matrix by inner product the input tensor.
	Loss function	cross-entropy	Cross entropy loss function
Hyperparameters of Spatial-ID	edge_weight	TRUE	Whether to use edge weight in spatial neighbor graph. Using edge weight means nearer neighbors will contribute more to results.
	k_graph	30	Number of neighbors in spatial neighbor graph.
	theta	15	Decay coefficient for edge weight
	w_dae	1	Weight of autoencoder loss.
	w_gae	1	Weight of variational graph autoencoder loss.
	w_cls	10	Weight of the classifier loss for self-supervised learning.
	epochs	200	Number of training epochs.
New cell type discovery	Threshold	0.5-0.9	The threshold in the thresholding step to determine unassigned cells. For the mouse primary motor cortex dataset, the threshold is set as 0.9. For the human NSCLC dataset, the threshold is set as 0.7.
	Clusters	approximate half of total cell types	The unassigned cells are grouped in to the number of clusters in the clustering step.

Table 4 in revised manuscript

Line 380-383 in revised manuscript:

In addition, we introduce a set of experiments to compare the effect of spatial information (i.e., number of neighbor cells) on this dataset (Extend Fig. 7e). We can observe that the number of neighbor cells slightly affects the performance of accuracy, and the larger number of neighbor

cells may lead to reduced differences between local cells.

4. While code has been provided, there is very limited documentation and only analysis scripts rather than a module/package. Without additional packaging and documentation, I think it will be difficult for users to run this method on their own data.

Reply: Thanks for your suggestion. We add a demo for the open source code and more instruction on how to run this method, please refer to the github link. (<https://github.com/TencentAILabHealthcare/spatialID>)

Minor suggestions:

1. A little more elaboration of why we need Stage 3 after Stage 2 would be useful in the Results section. Why aren't the results from Stage 2 sufficient? Explain the intuition for why we need to use this self-supervised approach.

Reply: Thanks for your suggestion. Stage 1 involves knowledge transfer from reference datasets. The pretrained DNN in stage 1 is used to generate pseudo-labels for each cell in SRT dataset. Stage 2 involves feature embedding of gene expression and spatial information, and employs self-supervised strategy to train the classifier using the generated pseudo-labels in stage 1. The stage 1 and stage 2 perform the model training, the output of stage 2 is used to calculate loss in each epoch. The training process optimizes the parameters of the model until convergence, and saves the optimal model. In general, the optimal model is obtained after the training. Therefore, the stage 3 (inference stage) reloads the optimal model from stage 2 to output the final results.

Essentially, the self-supervised approach trains the GCN in stage 2 using the pseudo-labels that are generated from the gene expression profiles of the SRT dataset itself by the pretrained DNN and the temperature setting strategy.

Line 179 – 207 in manuscript:

The pipeline of our proposed Spatial-ID is shown in Fig. 1, including 3 stages. Stage1 involves knowledge transfer from reference datasets. Stage 2 involves feature embedding of gene expression profiles and spatial information of SRT dataset, and employs self-supervised strategy to train the classifier. Stage 3 uses the optimal model derived from stage 2 to perform cell type annotation for SRT dataset.

Formally, the DNN in stage 1 is trained on the gene expression profiles and ground truth labels of reference datasets (Fig. 1a). The pretrained DNN is used to generate the probability distribution for each cell of SRT dataset, then the probability distribution is used to construct pseudo-label (Fig. 1b) through a temperature setting strategy (See Methods). The GCN in stage 2 contains an autoencoder, a variational graph autoencoder and a classifier. Given a SRT dataset, the gene expression profiles are transformed into a cell-gene matrix (i.e., gene

expression matrix I), and a spatial neighbor graph (i.e., cell-cell adjacency matrix A) is constructed by considering each cell as a node and the spatial location relationships between cells as edges, where the relationship weight of each pair of cells is negatively associated with Euclidean distance (Fig. 1d). The autoencoder (Fig. 1c) takes the gene expression matrix and outputs the encoded gene representations X . The adjacency matrix A and the encoded gene representations X are fed into the variational graph autoencoder (Fig. 1e), that performs spatial embedding for the gene representations and outputs the final latent representations Z that provide comprehensive characters of gene expression profiles and spatial information. Thereafter, the final latent representations Z are used to reconstruct the gene expression matrix in the autoencoder and the adjacency matrix in the variational graph autoencoder, respectively. Simultaneously, the self-supervised learning strategy employs the final latent representations Z and the generated pseudo-labels L from stage 1 to train the classifier. In stage 2, the training process optimizes the parameters of the GCN model until convergence, and saves the optimal model. Finally, the stage 3 (i.e., inference stage) reloads the optimal model from stage 2, and output the cell type predictions of a given SRT dataset.

2. In the writeup of results for Fig. 4, the text does not introduce the panels in alphabetical order (e.g. 4e discussed before 4c). 4b is also not mentioned.

Reply: Thanks for your suggestion. We rewrite this part to make it clearer.

Line 399 – 419 in revised manuscript:

The annotation result of Spatial-ID is shown in Fig. 4b. Based on the quantitative comparison, Spatial-ID also demonstrates superior accuracy of cell type identification on the mouse spermatogenesis dataset (Fig. 4c). Specifically, Spatial-ID achieves mean accuracy 60.45% on all the 6 samples (Fig. 4c) and achieves mean weighted F1 score 0.55 (Extend Fig. 7b). The ablation analysis shows that the DNN achieves mean accuracy 58.27% on all the 6 samples (Extend Fig. 7d). This again illustrates the importance of spatial information. Notably, Cell2location presents excellent performance and achieves the best accuracy 62.88% on mouse spermatogenesis dataset. The Slide-seq technology provides an approximate cell-level spatial resolution, where each spot contains several cells (i.e., 1 to 10). Technically, this dataset is more suitable for deconvolution-based methods. Besides, the ground truth of this dataset is derived from a non-negative matrix factorization regression (NMF) method (Table 1), that may introduce method bias for ground truth labels. This may allow regression-based methods, such as Cell2location, to achieve better results. Nevertheless, Spatial-ID achieves comparable accuracy with Cell2location. These results suggest that Spatial-ID can also effectively handle the spot-based Slide-seq dataset with tens of thousands of genes, even if the area of a spot spans more than one cell. As analysis in previous study, diabetic induces testicular injuries through disrupting the spatial structures of seminiferous tubules and changing the expression pattern of many genes at molecular level. We can observe relatively regular spatial structure of seminiferous tubules in wild-type mouse (Fig. 4d), and irregular spatial structure of seminiferous tubules in diabetic mouse (Fig. 4e).

3. The description of the UMAP as separating cell types into “separable communities” seems a bit misleading given that most clusters aren’t actually separating.

Reply: Thanks for your comment. The word “separable” here is a bit misleading. What we mean here is that most identified cell types congregate into groups in the low dimension feature space, which can be distinguished from each other. We replace the statement of “separable” to the “distinguishable”, see the revision in the manuscript.

Line 485-487 in revised manuscript:

Besides, by color-coding our identified cell types in low dimension feature space, most identified cell types congregate into distinguishable communities.

4. Line 426, should introduce Hotspot (briefly) before talking about results from it.

Reply: Thanks for your suggestion. We briefly introduced Hotspot before reporting results.

Line 535-538 in revised manuscript:

Based on the notion that genes expressed at similar levels by proximal cells must be varying in an informative manner, Hotspot allows the identification of spatially informative genes and spatially dependent patterns of gene expression in situ for SRT datasets.

5. Line 668, this description is not clear to me. Maybe provide an example of how this is done.

Reply: Thanks for your suggestion. To generate simulated SRT datasets, we employ a random-gene-discard strategy to randomly reset part of values to 0 in the gene expression matrix. A simple example is shown below. The left table shows a gene expression matrix, the right table shows the simulated gene expression matrix with dropout rate 0.5 that a random half of values are set to 0.

0.281	0.113	0.625	0.290	0.590
0.095	0.488	0.515	0.121	0.059
0.833	0.692	0.572	0.001	0.018
0.779	0.884	0.011	0.633	0.434
0.160	0.557	0.263	0.837	0.124
0.470	0.461	0.859	0.169	0.623

0.281	0.113	0.625	0.000	0.590
0.095	0.000	0.000	0.121	0.000
0.000	0.692	0.000	0.000	0.000
0.779	0.000	0.000	0.633	0.434
0.000	0.557	0.263	0.000	0.000
0.000	0.461	0.000	0.169	0.623

Line 801-802 in revised manuscript:

The simulated SRT datasets are generated by a random-gene-discard strategy, that randomly resets part of values to 0 in the gene expression matrix.

Reviewer #2 (Remarks to the Author):

The authors present Spatial-ID a novel method for annotating cells or spots in cellular or

subcellular resolution spatial transcriptomic data (e.g. Stereo-seq) that combines: a deep neural network trained on a reference single-cell or single-nucleus RNAseq dataset which generates labels that are further refined using an autoencoder-driven embedding that combines the transcriptomic data from the spatial method with the spatial location of cells via a kNN graph. This is compared to several methods of reference-based autoannotation of single-cell or single-nucleus RNAseq data.

While the method is novel and shows improvement over the non-spatial methods, the authors do not compare their tool to existing tools designed to annotate and deconvolve spatial transcriptomics data. The manuscript also has insufficient description of their method to enable reproduction of their results particularly with respect to hyper-parameters.

Reply: Thanks for your valuable comments. We really appreciate your efforts in reviewing our manuscript. We have revised the manuscript accordingly. We add SOTA deconvolution-based annotation method in the benchmarking analysis. We add 2 table in revised manuscript to explain all hyperparameters and professional terms in detail. Our point-by-point responses for all comments are detailed below. The revised portion are marked in red in each response.

Major :

Spatial ID is much more similar in concept to spatial deconvolution methods - using a well annotated reference single-cell dataset to label spatial transcriptomics data either with a single label per spot or with a set of cell-type proportions per spot - but the introduction primarily discusses single-cell cell-type annotation tools. Likewise, it should be compared to spatial deconvolution tools in addition to or instead of single-cell automatic annotation tools.

Reply: Thanks for your suggestion. We add a leading deconvolution method, e.g., Cell2location, in the benchmarking analysis and report the performance of Cell2location in the revised manuscript.

Although the emerging SRT technologies, such as MERFISH, Slide-seq and Stereo-seq, have achieved cell-level or subcellular spatial resolution, several deconvolution-based methods, such as Cell2location, can also support cell-type annotation at cell-level spatial resolution. Therefore, we choose Cell2location as the representative deconvolution-based method for benchmark analysis.

For the Cell2location method, the accuracy of each sample from the 4 datasets is shown in the following tables, that are also be visualized for comparison in Fig. 2f, Fig. 3e, Fig. 4c, Fig. 5a. Moreover, the comparison of weighted F1 score can be found in Extend Fig. 2a, Extend Fig. 7a, Extend Fig. 7b, Extend Fig. 7c. The comparison of runtime efficiency can be found in Table 3. The robustness test of Cell2location can be found in Fig. 2j and Extend Fig. 3.

We can observe that Cell2location presents excellent performance. Especially,

Cell2location can achieve SOTA performance on the mouse spermatogenesis dataset measured by Slide-seq. The Slide-seq technology provides an approximate cell-level spatial resolution, where each spot contains several cells (1 to 10). Technically, this dataset is more suitable for deconvolution-based methods. Our proposed Spatial-ID achieves comparable performance in accuracy. Besides, the runtime efficiency of Spatial-ID is much better than Cell2location (Fig .4f, Table 3 in the revised manuscript).

Sample	Spatial-ID	Cell2location
mouse1_sample1	91.20%	90.66%
mouse1_sample2	92.08%	92.58%
mouse1_sample3	92.92%	92.23%
mouse1_sample4	91.36%	92.34%
mouse1_sample5	92.26%	92.88%
mouse1_sample6	92.99%	92.70%
mouse2_sample1	92.85%	92.68%
mouse2_sample2	93.66%	93.50%
mouse2_sample3	93.58%	92.80%
mouse2_sample4	93.43%	93.52%
mouse2_sample5	93.25%	92.74%
mouse2_sample6	93.42%	93.20%

Table 2-1 Mouse brain - primary motor cortex (MOP) dataset. Mean accuracy for Spatial-ID and Cell2location: 92.75%, 92.65%.

Sample	Spatial-ID	Cell2location
sample1	86.94%	80.81%
sample2	87.51%	79.81%
sample3	88.76%	78.50%

Table 2-2 Mouse brain – hypothalamic preoptic region (3D) dataset. Mean accuracy for Spatial-ID and Cell2location: 87.74%, 79.71%.

Sample	Spatial-ID	Cell2location
diabetes_1	59.66%	59.63%
diabetes_2	59.26%	63.15%
diabetes_3	62.84%	65.05%
wild_type_1	55.66%	58.81%
wild_type_2	55.61%	61.62%
wild_type_3	69.82%	69.03%

Table 2-3 Mouse spermatogenesis dataset. Mean accuracy for Spatial-ID and Cell2location: 60.45%, 62.88%.

Sample	Spatial-ID	Cell2location
1	37.32%	48.92%
2	48.91%	58.49%
3	35.29%	48.92%
4	36.59%	55.32%
5	94.22%	69.78%
6	72.04%	69.84%
7	52.86%	64.36%
8	68.80%	69.22%
9	87.83%	71.43%
10	74.50%	71.41%
11	62.08%	63.00%
12	81.23%	70.06%
13	81.29%	76.72%
14	86.91%	80.27%
15	74.90%	64.64%
16	77.58%	67.67%
17	82.38%	68.70%
18	86.15%	62.74%
19	66.95%	54.04%
20	87.08%	63.55%

Table 2-4 Human NSCLC (non-small-cell lung cancer) dataset. Mean accuracy for Spatial-ID and Cell2location: 69.76%, 64.95%.

Fig. 4f Runtime per sample on the mouse spermatogenesis dataset

Fig. 2f and Fig. 3e show the accuracy performance for mouse primary motor cortex (MOP) dataset and mouse hypothalamic preoptic region (3D) dataset.

Fig. 4c and Fig. 5a show the accuracy performance for mouse spermatogenesis dataset and human NSCLC dataset.

Extend Fig. 2a and Extend Fig. 7a show the weighted F1 score for mouse primary motor cortex (MOP) dataset and mouse hypothalamic preoptic region (3D) dataset.

Extend Fig. 7b and Extend Fig. 7c show the weighted F1 score for mouse spermatogenesis dataset and human NSCLC dataset.

Fig. 2j and Extend Fig. 5a show for robustness test of Spatial-ID and the control methods on the mouse primary motor cortex (MOP) dataset. Left: accuracy; Right: weighted F1 score.

We also add a description of Cell2location in the introduction section as follow.

Line 115 to 124 in revised manuscript:

The deconvolution-based methods were a kind of primary integrated analysis methods, such as SPOTlight, RCTD and Cell2location, that learned cell-type gene signatures from reference datasets to disentangle discrete cellular subpopulations from mixtures of mRNA transcripts from each capture spot in SRT datasets. Although the emerging SRT technologies, such as MERFISH, Slide-seq and Stereo-seq, had achieved cell-level or subcellular spatial resolution, some deconvolution-based methods, e.g., Cell2location, can still support cell-type annotation at cell-level spatial resolution. Specifically, Cell2location employed an interpretable hierarchical Bayesian model to map the spatial distribution of cell types by integrating scRNA-seq data and multi-cell SRT data from a same tissue.

Spatial-ID claims the improvement in its accuracy is derived from inclusion of spatial location information, yet this is incorporated only in a small portion of the overall model. How much of an improvement in accuracy is truly derived from the addition of spatial location? Could the authors repeat their benchmarks with their method excluding the adjacency matrix to quantify the benefits of incorporating spatial location information?

Reply: Thanks for your comment. To verify the spatial information, we introduce an ablation analysis that excludes the spatial information. Without the spatial information, the GCN in the stage 2 does not work and the framework is effectively a DNN model. Therefore, the ablation analysis compares the results of Spatial-ID with the results of DNN (Stage 1). The result is shown in Extend Fig. 7d.

Extend Fig. 7d ablation analysis on the 4 public datasets. MOP: mouse primary motor cortex (MOP) dataset. HPR: mouse hypothalamic preoptic region dataset. Testis: mouse spermatogenesis dataset. NSCLC: human non-small-cell lung cancer dataset.

Line 270-273 in revised manuscript:

If we abandon the spatial information, the DNN can achieves mean accuracy 91.96% on all the 12 samples (Extend Fig. 7d). Obviously, this ablation analysis demonstrates that it is beneficial to use spatial information in Spatial-ID.

Line 370-372 in revised manuscript:

The DNN can achieves mean accuracy 85.00% on all the 3 samples (Extend Fig. 7d). Obviously, spatial location information is also beneficial for Spatial-ID on the 3D SRT dataset.

Line 259-261 in revised manuscript:

The ablation analysis shows that the DNN achieves mean accuracy 58.27% on all the 6 samples (Extend Fig. 7d). This again illustrates the importance of spatial information.

Line 403-404 in revised manuscript:

The ablation analysis shows that the DNN achieves mean accuracy 68.09% on all 20 samples (Extend Fig. 7d).

Results:

It's unclear where the ground truth for the cell/spot labelling came from that the methods were benchmarked against. Were the author reported computationally-derived annotations used? Was it manually annotated?

Reply: Thanks for your comment. This is a valid point. All the ground truth labels we used were provided with the data of the original research papers (See Table 1 in revised manuscript). In these papers, these cell type labels were annotated by combining computational analysis (e.g., clustering analysis or matrix factorization based analysis) and manual labelling of biologist and histologist. It is the involvement of manual labelling that incorporated the established knowledge and guided the computational analysis, making the given label reliable. Such practice is the most widely accepted standard in cell type annotation. The established knowledge includes known marker genes of specific cell types and known tissue structures inferred by the spatial context. Spatial-ID's purpose is to annotate the cell types as close as possible to these semi-manual labels. Our benchmarking analysis have proved that Spatial-ID did this job better than other computational methods.

We summarize the source of labels for all the datasets used in this work.

Datasets	Source of labels
Mouse brain - primary motor cortex (MOP)	cluster analysis and manual labelling
snRNA-seq 10x v3 B	cluster analysis and manual labelling
Mouse brain - hypothalamic preoptic region (3D)	cluster analysis and manual labelling
GSE113576	cluster analysis and manual labelling
Mouse spermatogenesis	non-negative matrix factorization regression (NMFreg)-based statistical method
GSE112393	Based on marker genes manually
Human NSCLC - non-small-cell lung cancer	Deconvolution-based method and cluster analysis
scRNA-seq NSCLC	cluster analysis and manual labelling
Mouse brain atlas of cell types from the Linnarsson Lab	cluster analysis and manual labelling
Mouse brain hemisphere (3D)	None

The public datasets have provided ground truth labels for cells/spots. According to the documents and papers of these datasets, we summarize the source of their labels. The mouse brain datasets employ cluster analysis to construct cell type taxonomy. It should be noted that the cluster analysis usually involves statistical assessments to remove low-quality clusters or manually filter outliers. Therefore, the datasets with labels derived from cluster analysis have more distinguishable characteristics in the feature space.

Line 645-655 in revised manuscript:

All the ground truth labels we used were provided with the data of the original research papers (See Table 1). In these papers, these cell type labels were annotated by combining computational analysis (e.g., clustering analysis or matrix factorization based analysis) and manual labelling of biologist and histologist. It is the involvement of manual labelling that incorporated the established knowledge and guided the computational analysis, making the given label reliable. For example, the mouse primary motor cortex dataset that involves clustering analysis and manually filtering low-quality or doublet-driven clusters, shows more distinguishable characteristic in the feature space. Such practice is the most widely accepted standard in cell type annotation. The established knowledge includes known marker genes of specific cell types and known tissue structures inferred by the spatial context.

It should be made clear throughout the manuscript that a distinct pipeline is necessary for novel cell-type discovery that is not inherent to the Spatial-ID pipeline (Figure 1 k). In the introduction and abstract it is made to sound like Spatial-ID simultaneously finds novel cell-types and identified reference cell-types but this is not the case. In addition, the authors should comment on the fact that the novel cell type pipeline identified 8 novel clusters when then expected only 2 - could they identify the remaining 6 novel clusters as truly novel cell-types as well?

Reply: Thanks for your comment. To make the process of new cell type discovery clearer, we describe more details of the analysis in the manuscript. Besides, we add a Human NSCLC (non-small-cell lung cancer) dataset and perform the process of new cell type discovery.

Essentially, the new cell type discovery is performed on the cell type probability distribution outputted by cell typing methods. It is a postprocess of cell typing methods that attempts to identify new categories from low-confidence cells.

As shown in Fig 2k, the process of new cell type discovery contains Thresholding, Clustering and Mapping & Filtering sub-steps.

(5) Thresholding: the maximum score $c_s \in [0, 1]$ of predicted cell type probability distribution of each cell is first examined by a threshold. If the maximum score

c_s of a cell is less than the threshold, the cell is tagged as an unassigned cell. The lower threshold retrieves less unassigned cells, that makes the subsequent new class discovery process impossible to find new cell type patterns. On the other hand, the higher threshold retrieves more unassigned cells, that leads more clusters to filter in the subsequent analysis. The output layer of classifier in Spatial-ID is a softmax function, therefore the cells that are easier to classify usually have a maximum score close to 1. The threshold is empirically set in the range of 0.5 to 0.9. We set the threshold of 0.9 for demonstration in Fig. 2k.

- (6) Clustering: The clustering step is performed to group the unassigned cells into different clusters. Empirically, the number of clusters approximately equals to the half of total cell types. For example, the reference dataset contains 20 cell types. In Fig. 2k, the unassigned cells are clustered into 9 clusters.
- (7) Mapping: According to the clusters, the mapping step aligns the clusters of unassigned cells to the predicted cell types in the same feature space, as shown in Extend Fig. 2g. Compared with the prediction of Spatial-ID (Extend Fig. 2f), each cluster contains several types predicted by Spatial-ID. For example, the cluster 2 contains the L2/3 IT and L5 IT cell types predicted by Spatial-ID, the cluster 5 contains the L6 IT cell type predicted by Spatial-ID.
- (8) Filtering: The distance from the center of each cluster to the center of the predicted types is analyzed. If the distance is less than the radius of one predicted type in the feature space, the cluster is still identified as this predicted type rather than a new cell type. If the distance is larger than the radius of all predicted types in feature space, the cluster is identified as a new cell type. For example, the distance of cluster 2 to L2/3 IT cell types predicted by Spatial-ID in feature space is larger than the radius of predicted L2/3 IT cell types, as well as L5 IT cell types predicted by Spatial-ID. Therefore, the cluster 2 is identified as a new cell type. This new cell type is labelled as L4/5 IT in ground truth (Extend Fig. 2e). Compared with the ground truth, we can calculate the recall 0.258, precise 0.945 and F1 score 0.405 for this new cell type. Similarly, the cluster 5 is identified as a new cell type, that is labelled as L6 IT car3 in ground truth. Apply the above filtering rule to the other clusters (0,1,3,4,6,7,8) in turn, we can find that these clusters are closer to their main cell type predicted by Spatial-ID, therefore these clusters are not identified as new cell types. For example, the distance of cluster 0 to L6 IT is too closer, so this cluster is still identified as L6 IT. The distance of cluster 4 to L2/3 IT is too closer, so this cluster is still identified as L2/3 IT.

Fig. 2k

Extend Fig. 2f

Extend Fig. 2g

For the Human NSCLC (non-small-cell lung cancer) dataset, we also perform new cell type discovery after Spatial-ID, as shown in Fig. 5e-h. The process of new cell type discovery can retrieve the cell types of Neutrophil and Endothelial that are missed in the prediction of Spatial-ID (Fig .5d).

Fig. 5e-h

Line 320 – 352 in revised manuscript for MOP dataset:

L4/5 IT and L6 IT Car3 neurons are not annotated in the snRNA-seq 10x v3 B dataset, thus Spatial-ID predicts the L6 IT Car3 neurons as L6 IT neurons, and L4/5 IT neurons as L2/3 IT and L5 IT neurons (Fig. 2c and Fig. 2e). The ground truth of MOP dataset shows that L4/5 IT neurons populate in the continuous region between L2/3 IT and L5 IT neurons (Fig. 2d), and L6 IT Car3 neurons populate near the L6 IT neurons at L6 layer (Fig. 2d). These results suggest that the gene expression profiles of L4/5 IT neurons present gradual transition between the L2/3 IT and L5 IT neurons, and the characteristics of L6 IT Car3 neurons are similar to the L6 IT neurons (Fig. 2c). We conduct a postprocess for Spatial-ID to further distinguish these new cell types that are presented in the MOP dataset but unseen in the snRNA-seq 10x v3 B dataset. Based on the predictions of Spatial-ID, a pipeline of new cell type discovery (Fig. 2k) is performed to determine whether there are new cell types in the MOP dataset, including thresholding, clustering, mapping and filtering (See Methods). The unassigned cells are determined by a threshold (i.e. 0.9) at the thresholding step. The clustering step is performed to group the unassigned cells into different clusters (i.e. 9 clusters). According to the clusters, the mapping step aligns the clusters of unassigned cells to predicted cell type in the same feature space, as shown in Extend Fig. 2g. Compared with the predictions of Spatial-ID (Extend Fig. 2f), each cluster contains several cell types predicted by Spatial-ID. For example, the cluster 2 contains the L2/3 IT and L5 IT cell types predicted by Spatial-ID, the cluster 5 contains the L6 IT cell type predicted by Spatial-ID. The filtering step analyzes the distance from the center of each cluster to the center of the predicted cell types. If the distance is less than the radius of one predicted cell type in feature space, the cluster is still identified as this predicted cell type rather than a new cell type. If the distance is larger than the radius of all predicted cell types in the feature space, the cluster is identified as a new cell type. For example, the distance of cluster 2 to L2/3 IT cell types predicted by Spatial-ID in feature space is larger than the radius of predicted L2/3 IT cell types, as well as L5 IT cell types predicted by Spatial-ID (Extend Fig. 2f and Extend Fig. 2g). Therefore, the cluster 2 is identified as a new cell type that is labelled as L4/5 IT in ground truth (Extend Fig. 2e). Similarly, the cluster 5 is identified as a new cell type, that is labelled as L6 IT car3 in ground truth (Extend Fig. 2e). Applying the above filtering rule to the other clusters in turn, we can find that these clusters are closer to their main cell type predicted by Spatial-ID, therefore these clusters are not identified as new cell types. Finally, the discovered L4/5 IT and L6 IT Car3 neurons achieves F1 score of 0.405 and 0.904, respectively (Fig. 2k).

Line 454 – 460 in revised manuscript for Human NSCLC dataset:

The thresholding step determines a set of unassigned cells by a threshold 0.7 (Fig. 5f). Then, these unassigned cells are grouped into 5 clusters at the clustering step (Fig. 5g). Next, the mapping step aligns the clusters to predicted cell types in the feature space for these unassigned cells (Fig. 5g and Fig. 5e). Finally, the filtering step analyzes the distance from the center of each cluster to the center of the predicted cell types. The cluster 0 and cluster 4 are identified as new cell types (Fig. 5h). Obviously, they are labelled as neutrophil cells and endothelial cells in ground truth.

Line 806-840 in Methods section of revised manuscript:

The discovery of new cell populations has biologically important implication for omics analysis. Technically, the new cell type discovery problem can be characterized as an anomaly detection task. For example, the reference datasets do not contain the new cell types to be found in SRT datasets or have serious class imbalance problem derived from scarce cells of the new cell types, thus these new cell types are identified into incorrect cell types (e.g., L4/5 IT and L6 IT car3 in mouse primary motor cortex dataset). For another situation, the SRT datasets have indistinguishable feature distributions (e.g., the indistinguishable cell types in human NSCLC dataset).

Essentially, the new cell type discovery task is performed on the cell type probability distribution outputted by cell typing methods. It is a postprocess of cell typing methods that attempts to identify new categories from low-confidence cells. As shown in Fig 2k, the process of new cell type discovery contains thresholding, clustering, mapping and filtering sub-steps. For each cell, the maximum score $c_s \in [0,1]$ of predicted probability vector $C = \{c_1, \dots, c_i, \dots, c_{n-1}\}$, $\sum_1^{n-1} c_i = 1$ is first examined by a threshold at the thresholding step. If the maximum score c_s of the cell is less than the threshold (e.g., 0.9 in Fig. 2k), the cell is tagged as an unassigned cell. The lower threshold retrieves fewer unassigned cells, that makes the subsequent new class discovery process impossible to find new cell type patterns. On the other hand, the higher threshold retrieves more unassigned cells, that leads more clusters to filter in the subsequent analysis. The threshold is empirically set in the range of 0.5 to 0.9. We set a threshold of 0.9 for demonstration in Fig. 2k and a threshold of 0.7 for demonstration in Fig. 5e-h. The clustering step is performed to group the unassigned cells into different clusters, that could ensure that the new cell types populate in a relatively isolated area in the feature space. Empirically, the number of clusters approximately equals to the half of total cell types. Next, the mapping step aligns the clusters to predicted cell types in the feature space for these unassigned cells (Extend Fig. 2g and Fig. 5e-g). Each cluster may contain multiple cell types predicted by Spatial-ID. For example, the cluster 2 in Extend Fig. 2g contains the L2/3 IT and L5 IT cell types predicted by Spatial-ID, and the gene expression profile of this cluster presents gradual transition between the L2/3 IT and L5 IT cell types. The filtering step analyzes the distance from the center of each cluster to the center of the predicted types one-by-one. If the distance is less than the radius of one predicted type in feature space, the cluster is still identified as this predicted type rather than a new cell type. If the distance is larger than the radius of all predicted types in the feature space, the cluster is identified as a new cell type.

Methods:

There is a lack of specifics of the "novel cell type pipeline" - how the threshold for unassigned is determined, what clustering method was applied and what the parameters were set at and to what representation of the data the clustering was applied. The methods must be updated to include all of this information so that others can reproduce the author's work.

Reply: Thanks for your comment. We have comprehensively explained the process of novel cell type discovery in the last comment. The parameters of novel cell type discovery contain 'threshold' and 'clusters' shown in Table 4.

	Parameters	Value	Explanation
Data preprocess	cell_min_counts	300 (mouse brain hemisphere dataset)	Cells with total counts fewer than cell_min_counts will be removed.
	cell_max_counts_pct	98% (mouse brain hemisphere dataset)	Cells with total counts higher than cell_max_counts_pct quantile will be removed.
	filter_mt	10% (mouse brain hemisphere dataset)	Cells with percentage of mitochondrial genes larger than 10% will be removed.
	gene_min_cells	10 (mouse brain hemisphere dataset)	Genes presented in fewer than 10 cells will be removed.
DNN	Fully connected layer	4	Each fully connect layer is followed by a GRLU layer and dropout layer. Neurons in a fully connected layer have connections to all neurons in the previous layer. 4 fully connect layers contain 128, 64, 64, cell_types neurons for the mouse primary motor cortex dataset, mouse hypothalamic preoptic region dataset and human NSCLC dataset. 4 fully connect layers contain 512, 256, 256, cell_types neurons for the mouse spermatogenesis dataset and mouse brain hemisphere dataset.
	GELU layer	4	A type of activation function layer.
	Dropout layer	4	At training stage, individual neurons are either "dropped out" of the network (ignored) with probability p . Dropout layer is used to reduce overfitting.
	Loss function	Focal Loss	A modified cross entropy loss function is used to alleviate the class imbalance problem.
Autoencoder	pca_dim	None	PCA dimension. For the mouse spermatogenesis dataset and the mouse brain hemisphere dataset, this parameter is set as 200. Other datasets use all the shared genes.
	feat_dim	64	The encoded feature dimension.
	Fully connected layer	2	2 fully connect layers contain 100, 64 neurons, respectively. Each fully connect layer is followed by a batch normal layer, a ELU layer and a dropout layer.
	Batch normal layer	2	Batch normal layer is used to make training of neural networks faster and more stable through normalization of the layers' inputs by re-centering and re-scaling.
	ELU layer	2	A type of activation function layer.
	Dropout layer	2	As above of dropout layer.
	Decoder layer	1	The decoder layer is also a fully connected layer. Neurons in this layers equal to the input dimension (e.g., pca_dim or number of shared genes).
Variational graph autoencoder	sparse GCN layer	2	A GCN layer receives feature vector of each node and neighbor graph (adjacency matrix). GCN layers constrains 32, 8 neurons, respectively. Each GCN layer is followed by a ReLU layer and a dropout layer.
	RELU layer	2	A type of activation function layer.
	Dropout layer	2	As above of dropout layer.
	Inner-product layer	1	Inner-product layer is used to reconstruct the adjacency matrix by inner product the input tensor.
	Loss function	cross-entropy	Cross entropy loss function
Hyperparameters of Spatial-ID	edge_weight	TRUE	Whether to use edge weight in spatial neighbor graph. Using edge weight means nearer neighbors will contribute more to results.
	k_graph	30	Number of neighbors in spatial neighbor graph.
	theta	15	Decay coefficient for edge weight
	w_dae	1	Weight of autoencoder loss.
	w_gae	1	Weight of variational graph autoencoder loss.
	w_cls	10	Weight of the classifier loss for self-supervised learning.
	epochs	200	Number of training epochs.
New cell type discovery	Threshold	0.5-0.9	The threshold in the thresholding step to determine unassigned cells. For the mouse primary motor cortex dataset, the threshold is set as 0.9. For the human NSCLC dataset, the threshold is set as 0.7.
	Clusters	approximate half of total cell types	The unassigned cells are grouped in to the number of clusters in the clustering step.

Table 4 in the revised manuscript.

Performance of neural networks and autoencoders tend to be sensitive to their hyperparameters. A full list of all hyperparameters and their values for each dataset analyzed in the manuscript must be provided in the Methods section.

Reply: Thanks for your suggestion. We report more details of the parameters in the revised manuscript. We summarize the parameters of DNN, autoencoder, variational graph autoencoder and other hyperparameters in Table 4 as shown above. For all SRT datasets, most of the hyperparameters are the same. The different hyperparameters are also explained in the table.

For a new application, the hyperparameters of neural network architecture do not need to be tuned. Users only need to modify the input dimension of DNN and the input dimension of the autoencoder (If PCA was applied, the input dimension should be set as the PCA dimension, e.g., 200). In the other hyperparameters, decay coefficient for edge weight is usually set as the pixels of 2 cell diameter in spatial. Users only need to modify the number of neighbor cells.

Minor:

Seurat v3 is misspelled & repeated at line 84-85

Reply: Thanks for your suggestion. We correct these typos.

"newly sequencing dataset" - sequenced at line 95, 99, 105

Reply: Thanks for your suggestion. We replace this phrase to “newly generated dataset”.

"PGM-based" - PGM needs to be defined.

Reply: Thanks for your suggestion. We rewrite the introduction of integrated analysis methods at line 116. In this paragraph, we highlight the integrated analysis methods related to cell type annotation, such as Tangram and Cell2location. The briefly introduction of BayesSpace was deleted because this PGM-based method was not used to cell type annotation.

PGM means probabilistic graphical model, that is a kind of machine learning algorithm. According to the references, we employ the term of integrated analysis methods to classify reference methods in this manuscript, thus the PGM term is deleted.

cell2location is a leading spatial deconvolution method that should be mentioned.

Reply: Thanks for your suggestion. We add this deconvolution-based method in the benchmarking analysis, and report the performance of Cell2location for the 4 public datasets in the manuscript.

Line 115 to 124 in revised manuscript:

The deconvolution-based methods were a kind of primary integrated analysis methods, such as SPOTlight, RCTD and Cell2location, that learned cell-type gene signatures from reference

datasets to disentangle discrete cellular subpopulations from mixtures of mRNA transcripts from each capture spot in SRT datasets. Although the emerging SRT technologies, such as MERFISH, Slide-seq and Stereo-seq, had achieved cell-level or subcellular spatial resolution, some deconvolution-based methods, e.g., Cell2location, can still support cell-type annotation at cell-level spatial resolution. Specifically, Cell2location employed an interpretable hierarchical Bayesian model to map the spatial distribution of cell types by integrating scRNA-seq data and multi-cell SRT data from a same tissue.

"self-collected Stereo-seq" (line 183) - did you mean newly generated Stereo-seq dataset?

Reply: Thanks for your suggestion. Yes, we collected this dataset ourselves that was generated by Stereo-seq technology. Please refer to the “Data collection of large field mouse brain hemisphere” in Method section for more details.

Introduction does not make it clear why the ability to cluster spatial transcriptomic data would imply spatial data would be useful for cell-type annotation. Description of the approach in the introduction (line 124-137) is unclear what the output of the method is and how the various algorithms arrive at that output - what dataset is it labelling? How are those labels arrived at?

Reply: Thanks for your suggestion. The description of Spatial-ID in introduction briefly introduces the important modules and techniques in Spatial-ID.

The pipeline of Spatial-ID in the Results section explains the process of cell type annotation for spatial data in detail.

Line 179 – 207 in revised manuscript:

The pipeline of our proposed Spatial-ID is shown in Fig. 1, including 3 stages. Stage1 involves knowledge transfer from reference datasets. Stage 2 involves feature embedding of gene expression profiles and spatial information of SRT dataset, and employs self-supervised strategy to train the classifier. Stage 3 uses the optimal model derived from stage 2 to perform cell type annotation for SRT dataset.

Formally, the DNN in stage 1 is trained on the gene expression profiles and ground truth labels of reference datasets (Fig. 1a). The pretrained DNN is used to generate the probability distribution for each cell of SRT dataset, then the probability distribution is used to construct pseudo-label (Fig. 1b) through a temperature setting strategy (See Methods). The GCN in stage 2 contains an autoencoder, a variational graph autoencoder and a classifier. Given a SRT dataset, the gene expression profiles are transformed into a cell-gene matrix (i.e., gene expression matrix I), and a spatial neighbor graph (i.e., cell-cell adjacency matrix A) is constructed by considering each cell as a node and the spatial location relationships between cells as edges, where the relationship weight of each pair of cells is negatively associated with Euclidean distance (Fig. 1d). The autoencoder (Fig. 1c) takes the gene expression matrix and outputs the encoded gene representations X . The adjacency matrix A and the encoded gene

representations X are fed into the variational graph autoencoder (Fig. 1e), that performs spatial embedding for the gene representations and outputs the final latent representations Z that provide comprehensive characters of gene expression profiles and spatial information. Thereafter, the final latent representations Z are used to reconstruct the gene expression matrix in the autoencoder and the adjacency matrix in the variational graph autoencoder, respectively. Simultaneously, the self-supervised learning strategy employs the final latent representations Z and the generated pseudo-labels L from stage 1 to train the classifier. In stage 2, the training process optimizes the parameters of the GCN model until convergence, and saves the optimal model. Finally, the stage 3 (i.e., inference stage) reloads the optimal model from stage 2, and output the cell type predictions of a given SRT dataset.

Reviewer #3 (Remarks to the Author):

This paper deals with the development of a method ("Spatial-ID") that can classify single cells analyzed in the context of spatially resolved transcriptomics (SRT). SRT is an important technique that does not only provide a snapshot of the gene expression status of a single cell (like ordinary scRNA-seq), but also provides information on the spatial location of cells relative to other cells analyzed.

The authors present a transfer learning based framework by which to determine the type of cells from SRT data sets. In more detail, the authors first train a deep neural network to generate labels (termed "pseudo-labels") using known reference data sets. Then, they classify the single cells from the SRT data set using the deep neural network, that is, they assign "pseudo-labels" to each single cell from the SRT data set. Subsequently, the pseudo-labels are used to train a graph neural network / autoencoder / variational graph autoencoder based neural network part that is concerned with learning appropriate embeddings and representations of the SRT data. When done training, the corresponding neural network part is used for final classification.

The theoretical part appears sound to me; however, the authors do not provide various details (definitions, parameters etc) that enable one to fully reproduce the architectures of the networks. Results demonstrate the superiority of Spatial-ID very convincingly. However, Results refer to mouse data sets alone, which has left me with some doubts. The analysis of additional data sets may be helpful to fully convince the reader of the universal benefits of the approach.

In summary, this appears to be a very promising paper to me, which however leaves a few important questions unanswered so far.

Reply: Thanks for your valuable comments. We really appreciate your efforts in reviewing our manuscript. We have revised the manuscript accordingly. We add 2 table in revised manuscript to explain all hyperparameters and professional terms in detail. To demonstrate the universality of our approach, we add a human NSCLC (non-small-cell lung cancer) dataset in the benchmarking analysis. Our point-by-

point responses for all comments are detailed below. The revised portions are marked in red in each response.

MAJOR:

* It would be appropriate to provide a concise, short summary of the open challenges remaining in spatially resolved transcriptomics, together with a description of advances of Spatial-ID that addresses the open challenges the authors have identified. The authors only list related work with the corresponding brief descriptions of the tools. Then they describe what Spatial-ID does. However, I am missing to put the deficiencies of earlier work into context with the advances Spatial-ID brings up. To be more specific: are there no methods that combine prior knowledge based on existing reference scRNA-seq datasets on the one hand, and spatial information retrieved from SRT datasets on the other hand?

Reply: Thanks for your comment. We review more related works in the introduction section. The integrated analysis methods combined SRT data with scRNA-seq data to bridge these trade-offs and provide a better understanding of the spatial organization of tissues. The representative methods, i.e., Tangram and Cell2location, are added in the benchmarking analysis. The integration analysis methods usually employ shared genes as anchors, and then build deconvolution or mapping relationships between the cells from the SRT data with scRNA-seq data. However, the available spatial information is also not efficiently used in these integrated analysis methods.

Until now, several spatial domains analysis methods, such as SpaGCN, SEDR, STAGATE, were proposed to spatial domain clustering analysis that enable coherent gene expression in spatial domains by integrating gene expression and spatial location together. But these methods did not integrate the prior knowledge from scRNA-seq datasets, and were not used for cell type annotation.

We also cite the following review works, that summarized the latest development in the single-cell omics field of integrated analysis.

“Argelaguet, Ricard, et al. "Computational principles and challenges in single-cell data integration." *Nature biotechnology* 39.10 (2021): 1202-1215.”

“Longo, Sophia K., et al. "Integrating single-cell and spatial transcriptomics to elucidate intercellular tissue dynamics." *Nature Reviews Genetics* 22.10 (2021): 627-644.”

Line 113 – 138 in revised manuscript:

By considering the characteristics of the SRT technologies, several integrated analysis methods combined SRT data with scRNA-seq data to bridge these trade-offs and provide a better understanding of the spatial organization of tissues. The deconvolution-based methods were a kind of primary integrated analysis methods, such as SPOTlight, RCTD and

Cell2location, that learned cell-type gene signatures from reference datasets to disentangle discrete cellular subpopulations from mixtures of mRNA transcripts from each capture spot in SRT datasets. Although the emerging SRT technologies, such as MERFISH, Slide-seq and Stereo-seq, had achieved cell-level or subcellular spatial resolution, some deconvolution-based methods, e.g., Cell2location, can still support cell-type annotation at cell-level spatial resolution. Specifically, Cell2location employed an interpretable hierarchical Bayesian model to map the spatial distribution of cell types by integrating scRNA-seq data and multi-cell SRT data from a same tissue. The mapping-based methods were another kind of primary integrated analysis methods that mapped each scRNA-seq cell to a specific spot or cell of SRT data from a same tissue. Specifically, Seurat v3 provided a comprehensive integration strategy to map scRNA-seq data and SRT data, and projected cellular states (e.g., cell type) from a reference dataset to newly generated datasets. Tangram employed a deep learning framework based on nonconvex optimization to align scRNA-seq data to various forms of spatial data collected from the same tissue, then mapped the cell types defined by scRNA-seq on the spatial context. However, these integrated analysis methods also did not efficiently use the available spatial information. Besides, several spatial domains analysis methods, such as SpaGCN, SEDR, STAGATE, were proposed to spatial domain clustering analysis that enable coherent gene expression in spatial domains by integrating gene expression and spatial location together. Inspired by the significant advancements of these spatial embedding-based clustering methods in identifying anatomical spatial domains, embedding spatial information should be beneficial to cell type annotation of SRT datasets.

This remains somewhat unclear.

* "the DNN model of transfer learning generates the probability distribution of each cell" -> which probability distribution do you mean? Please clearly define "the probability distribution of a cell".

Reply: Thanks for your comment. For example, on the mouse primary motor cortex (MOP) dataset, we train a DNN model with 20 cell types from the reference dataset. Then, we employ the trained DNN to predict each cell in the MOP dataset. Given the gene expression profile of a cell, the trained DNN model outputs a probability distribution of this cell to different cell types, where the probability distribution is a 20-dimensions vector, such as $V = [0.0, 0.0, 0.72, 0.04, 0.08, 0.02, 0.01, 0.01, 0.0, 0.0, 0.02, 0.01, 0.03, 0.02, 0.01, 0.0, 0.0, 0.0, 0.02, 0.01]$. According to the probability distribution, we can find the maximum value is 0.72 at index 2.

Table 5 in the revised manuscript lists the definition of machine learning terms.

Terms	Explanation
Transfer learning	The definition of transfer learning is that a machine learning model gains problem-solving knowledge from the source domain and stores the knowledge, then the model is applied to solve similar problems in the target domain.
Self-supervised learning	Self-supervised learning is a machine learning paradigm. It contains two steps. The first step generates the pseudo-labels. The second step, the actual task is performed with supervised learning using the pseudo-labels.
Temperature setting strategy	Temperature setting strategy adjusts the parameter called temperature in a standard softmax, then logit values of softmax are converted to pseudo-probabilities (i.e., pseudo-labels in our study), and higher values of temperature have the effect of generating a softer distribution of pseudo-probabilities among the output classes.
Pseudo-label	In contrast to the real labels or ground truth labels, the pseudo-labels are generated by an algorithm.
Fully connected layer	Fully connected layers connect every neuron in one layer to every neuron in another layer
Spatial embedding	Spatial embedding is one of feature learning techniques that allow complex spatial data to be used in neural networks and have been shown to improve performance in spatial analysis tasks.
Spatial neighbor graph	The spatial neighbor graph is a undirected graph defined for a set of points in a metric space, such as the Euclidean space.
Non-negative matrix factorization.	A method commonly used in bioinformatics for dimensionality reduction of gene expression data as the non-negativity constraint reflects that genes are either expressed or not and cannot be negatively expressed.
Probability distribution	In this study, a probability distribution is a mathematical description of the probabilities of cell types for a cell.
Euclidean distance	The Euclidean distance between two points in Euclidean space is the length of a line segment between the two points. It can be calculated from the coordinates of the points using the square root.
Autoencoder	An autoencoder is a type of artificial neural network used to learn efficient codings of input data, which is learned by attempting to regenerate the input from the encoded features.
GCN	A Graph convolutional network (GCN) is a class of neural networks for processing data that can be represented as graphs.
Variational graph autoencoder	Like a variational autoencoder, the input data of Variational graph autoencoder is sampled from a parametrized distribution, and the encoder and decoder are trained jointly such that the output minimizes a reconstruction error in the sense of the Kullback–Leibler divergence between the true posterior and its parametric approximation.

Table 5 in revised manuscript

* What is a "pseudo-label"? Please define/explain.

Reply: Thanks for your comment. Followed up on the previous comment, the probability distribution of each cell outputted by the trained DNN is converted to pseudo-label using the label smooth strategy. If $T=1$, the pseudo-label equal to the V in the previous comment. If $T=0$, the pseudo-label equal to $V_0=[0.0, 0.0, 1.0, 0.0, 0.0, 0.0, 0.0, 0.0, 0.0, 0.0, 0.0, 0.0, 0.0, 0.0, 0.0, 0.0, 0.0, 0.0, 0.0]$. If $T=\text{inf}$, the pseudo-label equal to $[0.05, 0.05, 0.05, 0.05, 0.05, 0.05, 0.05, 0.05, 0.05, 0.05, 0.05, 0.05, 0.05, 0.05, 0.05, 0.05, 0.05, 0.05, 0.05]$.

The definition this term can be found in Table 5 of the revised manuscript.

* What is a "temperature setting strategy of knowledge transfer"? Please explain, and provide suitable citations.

Reply: Thanks for your comment. Followed up on the previous comment, we have presented an example to explain the temperature setting strategy.

Formally, the pseudo-labels $L = \{l_1, \dots, l_i, \dots, l_n\}$ are derived from the output $Y = \{y_1, \dots, y_i, \dots, y_n\}$ of last fully connected layer of the DNN model by a modified *softmax* layer

$$l_i = \frac{e^{(y_i/T)}}{\sum_j e^{(y_j/T)}}$$

where T is the temperature parameter that is used to adjust the smoothness of distribution by the temperature setting strategy (Fig. 1b), and $L = D$, if $T = 1$. Obviously, a higher value for T produces softer distribution of pseudo-labels L over all classes.

To avoid ambiguity, we use the term of “temperature setting strategy” in the revised manuscript. This strategy is proposed in the follow citation, we have added this citation in the references.

«*Hinton, Geoffrey, Oriol Vinyals, and Jeff Dean. "Distilling the knowledge in a neural network." arXiv preprint arXiv:1503.02531 2.7 (2015).* »

The definition this term can be found in Table 5 of the revised manuscript.

Line 785 – 793 in revised manuscript:

Formally, the pseudo-labels $L = \{l_1, \dots, l_i, \dots, l_n\}$ are derived from the output $Y = \{y_1, \dots, y_i, \dots, y_n\}$ of last fully connected layer of the DNN model by a modified *softmax* layer

$$l_i = \frac{e^{(y_i/T)}}{\sum_j e^{(y_j/T)}}$$

where T is the temperature parameter⁰ that is used to adjust the smoothness of distribution by the temperature setting strategy (Fig. 1b), and $L = D$, if $T = 1$. Obviously, a higher value for T produces softer distribution of pseudo-labels L over all classes. The softer distribution of pseudo-labels L could transfers more information from the reference scRNA-seq datasets.

* How is the "Euclidean distance" between cells defined? In other words, what is the definition of a "cell" here?

Reply: Thanks for your comment. Every cell in SRT dataset has a coordinate in 2D or 3D space. The Euclidean distance between a pair of cells is calculated from their coordinate, such as $d = \sqrt{(x_1 - x_2)^2 + (y_1 - y_2)^2}$, where (x_1, y_1) and (x_2, y_2) are the coordinates of a pair cells. In general, the coordinate of a cell indicates its geometric center in space. For the mouse brain MOP dataset, the hypothalamic preoptic region dataset and the human NSCLC dataset, the outline of each cell is obtained by cell segmentation methods. For the mouse spermatogenesis dataset, we analyze the dataset at spot-wise, so the Euclidean distance is calculated from the coordinates of a pair of spots, which coordinates are determined by the sequencing array.

The definition this term can be found in Table 5 of the revised manuscript.

* All of the first paragraph after "The pipeline of Spatial-ID" is quite unclear for a reader, even if certainly knowledgeable in neural networks (which I am). Likely, a reader not knowledgeable in neural networks is entirely lost. It would be favorable to explain the workflow in more detail, for example, by making more accurate use of the figures that illustrate it. Also, one has to provide formal definitions for all concepts that are used in a formal context. In particular, what is a "cell", as an input for the neural networks in use here?

Reply: Thanks for your suggestion. We rewrite this part to make it clearer.

For each cell, its gene expression profile is a vector in the gene expression matrix I . In stage 1, the DNN deals the gene expression matrix I cell-by-cell, or batch-by-batch, where a batch contains multiple cells. In stage 2 and stage 3, the autoencoder and VGAE deal all the cells of SRT dataset in gene expression matrix I and spatial neighbor graph A .

Formally, we list the parameters of the network structure in Table 4.

Line 179 -207 in revised manuscript:

The pipeline of our proposed Spatial-ID is shown in Fig. 1, including 3 stages. Stage1 involves knowledge transfer from reference datasets. Stage 2 involves feature embedding of gene expression profiles and spatial information of SRT dataset, and employs self-supervised strategy to train the classifier. Stage 3 uses the optimal model derived from stage 2 to perform cell type annotation for SRT dataset.

Formally, the DNN in stage 1 is trained on the gene expression profiles and ground truth labels of reference datasets (Fig. 1a). The pretrained DNN is used to generate the probability distribution for each cell of SRT dataset, then the probability distribution is used to construct pseudo-label (Fig. 1b) through a temperature setting strategy (See Methods). The GCN in stage 2 contains an autoencoder, a variational graph autoencoder and a classifier. Given a SRT dataset, the gene expression profiles are transformed into a cell-gene matrix (i.e., gene expression matrix I), and a spatial neighbor graph (i.e., cell-cell adjacency matrix A) is constructed by considering each cell as a node and the spatial location relationships between cells as edges, where the relationship weight of each pair of cells is negatively associated with Euclidean distance (Fig. 1d). The autoencoder (Fig. 1c) takes the gene expression matrix and outputs the encoded gene representations X . The adjacency matrix A and the encoded gene representations X are fed into the variational graph autoencoder (Fig. 1e), that performs spatial embedding for the gene representations and outputs the final latent representations Z that provide comprehensive characters of gene expression profiles and spatial information. Thereafter, the final latent representations Z are used to reconstruct the gene expression matrix in the autoencoder and the adjacency matrix in the variational graph autoencoder, respectively. Simultaneously, the self-supervised learning strategy employs the final latent representations Z and the generated pseudo-labels L from stage 1 to train the classifier. In stage 2, the training process optimizes the parameters of the GCN model until convergence, and saves the optimal model. Finally, the stage 3 (i.e., inference stage) reloads the optimal

model from stage 2, and output the cell type predictions of a given SRT dataset.

* In Figure 1, what is 'CLS' supposed to mean? I was not able to find a definition throughout the entire text of the manuscript.

Reply: Thanks for your comment. The 'CLS' refers to the classifier.

* Is it possible to run Spatial-ID on a dataset that does *not* refer to mouse cells? The experiments provide the reader with the idea that the method predominantly works for mouse. Experiments on an additional species would be very convincing. If this is not possible, for example because there are no data available, please point this out and discuss the possible range of applications in the future.

Reply: Thanks for your comment. Apart from the 3 public datasets used in the original manuscript, we add a Human NSCLC (non-small-cell lung cancer) dataset in the benchmarking analysis. A new figure shows the results of benchmarking analysis and the new cell type discovery on this dataset.

Fig. 5 in the revised manuscript

Line 434-463 in revised manuscript:

We also perform a benchmarking analysis on a human non-small-cell lung cancer (NSCLC)

SRT dataset. We select the individual dataset Lung 9-1, including 20 samples and total 83,621 cells. These samples are generated from a Formalin-Fixed Paraffin-Embedded (FFPE) sample of a 60+ years old patient by high-plex spatial molecular imaging (0.18um per pixel) - CosMx SMI platform (www.nanostring.com). A panel of 980 genes are measured, and the field size of each sample is about 0.7mmx0.9mm. The reference scRNA-seq dataset (scRNA-seq NSCLC) includes 49,532 cells and 22,180 genes. The shared cell types contain mDC, Treg, fibroblast, T CD8, plasmablast, mast, T CD4, neutrophil, NK, macrophage, epithelial, pDC, endothelial, B-cell and tumors.

Fig. 5a shows the accuracy performance of Spatial-ID and the control methods, where Spatial-ID achieves the highest mean accuracy 69.76% on all 20 samples. Besides, Spatial-ID achieves mean weighted F1 score 0.6288 (Extend Fig. 7c). The ablation analysis shows that the DNN achieves mean accuracy 68.09% on all 20 samples (Extend Fig. 7d). The number of cells in some cell types of the human NSCLC dataset are scarce (Fig. 5b), and the ground truth labels of cells present indistinguishable communities in the feature space. We can observe that tumor cells predominate in this dataset (Fig. 5b, Fig. 5c). Affected by these factors, Spatial-ID misses some rare cell types, even though spatial-ID achieves the highest accuracy 87.73% on the example shown in Fig. 5d. To alleviate this problem, we employ the pipeline of new cell type discovery for the predictions of Spatial-ID (Fig. 5e). The thresholding step determines a set of unassigned cells by a threshold 0.7 (Fig. 5f). Then, these unassigned cells are grouped into 5 clusters at the clustering step (Fig. 5g). Next, the mapping step aligns the clusters to predicted cell types in the feature space for these unassigned cells (Fig. 5g and Fig. 5e). Finally, the filtering step analyzes the distance from the center of each cluster to the center of the predicted cell types. The cluster 0 and cluster 4 are identified as new cell types (Fig. 5h). Obviously, there are labelled as neutrophil cells and endothelial cells in ground truth. Although the postprocess of new cell type discovery can retrieve 2 missed cell types, there are still other missed cell types that are more scarce and indistinguishable. To further identify these cell types, more genes may need to be measured.

* Later, in Methods, the crucial details and parameters that describe the DNN, and also the GCN, are missing entirely. This means that the Methods remain irreproducible. Please provide a sufficiently detailed description of the architectures of all networks involved in the Methods part.

* Also, the descriptions of the autoencoders lack necessary details, such as sizes of layers, and so on. Maybe an appropriate figure can do the job.

Reply: Thanks for your suggestion. We summarize the parameters of DNN, autoencoder, variational graph autoencoder and other hyperparameters in Table 4.

	Parameters	Value	Explanation
Data preprocess	cell_min_counts	300 (mouse brain hemisphere dataset)	Cells with total counts fewer than cell_min_counts will be removed.
	cell_max_counts_pct	98% (mouse brain hemisphere dataset)	Cells with total counts higher than cell_max_counts_pct quantile will be removed.
	filter_mt	10% (mouse brain hemisphere dataset)	Cells with percentage of mitochondrial genes larger than 10% will be removed.
	gene_min_cells	10 (mouse brain hemisphere dataset)	Genes presented in fewer than 10 cells will be removed.
DNN	Fully connected layer	4	Each fully connect layer is followed by a GRLU layer and dropout layer. Neurons in a fully connected layer have connections to all neurons in the previous layer. 4 fully connect layers contain 128, 64, 64, cell_types neurons for the mouse primary motor cortex dataset, mouse hypothalamic preoptic region dataset and human NSCLC dataset. 4 fully connect layers contain 512, 256, 256, cell_types neurons for the mouse spermatogenesis dataset and mouse brain hemisphere dataset.
	GELU layer	4	A type of activation function layer.
	Dropout layer	4	At training stage, individual neurons are either "dropped out" of the network (ignored) with probability p . Dropout layer is used to reduce overfitting.
	Loss function	Focal Loss	A modified cross entropy loss function is used to alleviate the class imbalance problem.
Autoencoder	pca_dim	None	PCA dimension. For the mouse spermatogenesis dataset and the mouse brain hemisphere dataset, this parameter is set as 200. Other datasets use all the shared genes.
	feat_dim	64	The encoded feature dimension.
	Fully connected layer	2	2 fully connect layers contain 100, 64 neurons, respectively. Each fully connect layer is followed by a batch normal layer, a ELU layer and a dropout layer.
	Batch normal layer	2	Batch normal layer is used to make training of neural networks faster and more stable through normalization of the layers' inputs by re-centering and re-scaling.
	ELU layer	2	A type of activation function layer.
	Dropout layer	2	As above of dropout layer.
	Decoder layer	1	The decoder layer is also a fully connected layer. Neurons in this layers equal to the input dimension (e.g., pca_dim or number of shared genes).
Variational graph autoencoder	sparse GCN layer	2	A GCN layer receives feature vector of each node and neighbor graph (adjacency matrix). GCN layers constrains 32, 8 neurons, respectively. Each GCN layer is followed by a ReLU layer and a dropout layer.
	RELU layer	2	A type of activation function layer.
	Dropout layer	2	As above of dropout layer.
	Inner-product layer	1	Inner-product layer is used to reconstruct the adjacency matrix by inner product the input tensor.
	Loss function	cross-entropy	Cross entropy loss function
Hyperparameters of Spatial-ID	edge_weight	TRUE	Whether to use edge weight in spatial neighbor graph. Using edge weight means nearer neighbors will contribute more to results.
	k_graph	30	Number of neighbors in spatial neighbor graph.
	theta	15	Decay coefficient for edge weight
	w_dae	1	Weight of autoencoder loss.
	w_gae	1	Weight of variational graph autoencoder loss.
	w_cls	10	Weight of the classifier loss for self-supervised learning.
	epochs	200	Number of training epochs.
New cell type discovery	Threshold	0.5-0.9	The threshold in the thresholding step to determine unassigned cells. For the mouse primary motor cortex dataset, the threshold is set as 0.9. For the human NSCLC dataset, the threshold is set as 0.7.
	Clusters	approximate half of total cell types	The unassigned cells are grouped in to the number of clusters in the clustering step.

Table 4 in the revised manuscript.

* Referring to the definition of "pseudo-label" as well, I believe that is a good idea to provide a brief introduction to transfer learning, such that a reader not familiar with the basic concept can follow the paper. For example, just briefly point out what a "teacher" does, and what a "student" does. Then, everything should be clear.

Reply: Thanks for your suggestion.

The definition of transfer learning is that a machine learning model gains problem-solving knowledge from the source domain and stores the knowledge, then the model is applied to solve similar problems in the target domain. The source domain refers to the reference datasets, while the target domain refers to the SRT datasets. The DNN in stage 1 of Spatial-ID conforms to the conventional definition of transfer learning, thus does not contain a teacher model and a student model.

The teacher-student (T-S) learning is a special transfer learning paradigm, where a teacher model is used to “teach” a student model to make the same predictions as the teacher. Notably, the teacher model and the student model have different architectures. For example, the knowledge distillation employs the teacher-student (T-S) learning paradigm. “Hinton, Geoffrey, Oriol Vinyals, and Jeff Dean. "Distilling the knowledge in a neural network." arXiv preprint arXiv:1503.02531 2.7 (2015). “

To avoid ambiguity, we delete the sentence “In this strategy, the DNN model is the teacher model and the classifier is the student model.” in the self-supervised learning part. We add a sentence to explain the mechanism of self-supervised learning “Essentially, the self-supervised learning employs the labels that are generated using the gene expression features of the SRT dataset itself.”

The definition these terms can be found in Table 5 of the revised manuscript.

Terms	Explanation
Transfer learning	The definition of transfer learning is that a machine learning model gains problem-solving knowledge from the source domain and stores the knowledge, then the model is applied to solve similar problems in the target domain.
Self-supervised learning	Self-supervised learning is a machine learning paradigm. It contains two steps. The first step generates the pseudo-labels. The second step, the actual task is performed with supervised learning using the pseudo-labels.
Temperature setting strategy	Temperature setting strategy adjusts the parameter called temperature in a standard softmax, then logit values of softmax are converted to pseudo-probabilities (i.e., pseudo-labels in our study), and higher values of temperature have the effect of generating a softer distribution of pseudo-probabilities among the output classes.
Pseudo-label	In contrast to the real labels or ground truth labels, the pseudo-labels are generated by an algorithm.
Fully connected layer	Fully connected layers connect every neuron in one layer to every neuron in another layer
Spatial embedding	Spatial embedding is one of feature learning techniques that allow complex spatial data to be used in neural networks and have been shown to improve performance in spatial analysis tasks.
Spatial neighbor graph	The spatial neighbor graph is a undirected graph defined for a set of points in a metric space, such as the Euclidean space.
Non-negative matrix factorization.	A method commonly used in bioinformatics for dimensionality reduction of gene expression data as the non-negativity constraint reflects that genes are either expressed or not and cannot be negatively expressed.
Probability distribution	In this study, a probability distribution is a mathematical description of the probabilities of cell types for a cell.
Euclidean distance	The Euclidean distance between two points in Euclidean space is the length of a line segment between the two points. It can be calculated from the coordinates of the points using the square root.
Autoencoder	An autoencoder is a type of artificial neural network used to learn efficient codings of input data, which is learned by attempting to regenerate the input from the encoded features.
GCN	A Graph convolutional network (GCN) is a class of neural networks for processing data that can be represented as graphs.
Variational graph autoencoder	Like a variational autoencoder, the input data of Variational graph autoencoder is sampled from a parametrized distribution, and the encoder and decoder are trained jointly such that the output minimizes a reconstruction error in the sense of the Kullback–Leibler divergence between the true posterior and its parametric approximation.

Table 5 in revised manuscript

Line 704-707 in the revised manuscript:

The definition of transfer learning is that a machine learning model gains problem-solving knowledge from the source domain and stores the knowledge, then the model is applied to solve similar problems in the target domain. The source domain refers to the reference datasets, while the target domain refers to the SRT datasets.

Line 783-785 in the revised manuscript:

Essentially, the self-supervised learning employs the pseudo-labels that are generated using the gene expression features of the SRT dataset itself.

MINOR:

* Seruat -> Seurat

Reply: Thanks for your suggestion. We correct these typos.

* There are some minor edits to be done with respect to language, such as missing articles, and similar things. This is not so important, because I find the manuscript already perfectly readable.

Reply: Thanks for your suggestion. We review the full manuscript carefully, and correct some problems such as grammar issues. Besides, corresponding to the

modification, we add some references.

REVIEWER COMMENTS

Reviewer #1 (Remarks to the Author):

The revisions the authors made, particularly the inclusion of the NSCLC analysis and more explanation of the semi-supervised approach, have improved the manuscript. I have no further comments.

Reviewer #2 (Remarks to the Author):

The authors have addressed all of the points I raised, and the manuscript is much improved.

Reviewer #3 (Remarks to the Author):

The authors did an excellent job in addressing my comments.

Dear Reviewers:

On behalf of my co-authors, we gratefully thank all reviewers for reading our manuscript carefully and giving the positive comments.

REVIEWER COMMENTS

Reviewer #1 (Remarks to the Author):

The revisions the authors made, particularly the inclusion of the NSCLC analysis and more explanation of the semi-supervised approach, have improved the manuscript.

I have no further comments.

Reply: Thanks for your positive comments. We really appreciate your efforts in reviewing our manuscript again.

Reviewer #2 (Remarks to the Author):

The authors have addressed all of the points I raised, and the manuscript is much improved.

Reply: Thanks for your positive comments. We really appreciate your efforts in reviewing our manuscript again.

Reviewer #3 (Remarks to the Author):

The authors did an excellent job in addressing my comments.

Reply: Thanks for your positive comments. We really appreciate your efforts in reviewing our manuscript again.